# HyperHELM: Hyperbolic Hierarchy Encoding for mRNA Language Modeling

## Abstract

Language models are increasingly applied to biological sequences like proteins and mRNA, yet their default Euclidean geometry may mismatch the hierarchical structures inherent to biological data. While hyperbolic geometry provides a better alternative for accommodating hierarchical data, it has yet to find a way into language modeling for mRNA sequences. In this work, we introduce HyperHELM, a framework that implements masked language model pre-training in hyperbolic space for coding regions of mRNA sequences. Using a hybrid design with hyperbolic layers atop Euclidean backbone, HyperHELM aligns learned representations with the biological hierarchy defined by the relationship between mRNA and amino acids. Across multiple multi-species datasets, it outperforms Euclidean baselines on 9 out of 10 tasks involving property prediction, with 10% improvement on average, and excels in out-of-distribution generalization to long and low-GC content sequences; for antibody region annotation, it surpasses hierarchy-aware Euclidean models by 3% in annotation accuracy. Our results highlight hyperbolic geometry as an effective inductive bias for hierarchical language modeling of the CDS regions of mRNA sequences.

## 1 Introduction

Language models have been increasingly applied to biological sequence data, fueled by the growth of large-scale omics datasets (Lin et al., 2023; Celaj et al., 2023; Brixi et al., 2025). While originally designed for natural language, these models demonstrate promising performance in capturing dependencies within DNA (Zhou et al., 2024; Nguyen et al., 2024b;a; Brixi et al., 2025), RNA (Celaj et al., 2023; Prakash et al., 2024; Yazdani-Jahromi et al., 2025a;b), and protein sequences (Lin et al., 2023; Ferruz et al., 2022). The biological sequences, however, are structured differently from natural language, particularly in their hierarchical organization, where nucleotides or amino acids form motifs that can be nested within larger functional groups (Buhr et al., 2016). In this work, we take the rapidly expanding therapeutic domain of RNA, where the codon–amino acid hierarchy plays a key role in determining the biophysical properties of mRNA sequences and their expressed proteins (Clancy & Brown, 2008), and we focus on encoding this hierarchy directly into the representation space of a bio-language model by leveraging hyperbolic geometry.

While standard language models rely on Euclidean geometry, the number of concepts in hierarchies grows exponentially, outpacing the polynomial expansion of Euclidean volumes (Matoušek, 1996; 1999). This can severely limit the representation capacity of a model and hinder generalization (Liu et al., 2020). In contrast, the volume of hyperbolic space expands exponentially, maintaining well-separated representations across different branches of the hierarchy and reducing distortion in hierarchical relationships. The advantages of hyperbolic geometry are demonstrated in graph representation learning (Chami et al., 2019) and computer vision (Mettes et al., 2024), and are beginning to inform natural language modeling (He et al., 2024; 2025), though they have yet to be systematically applied to mRNA data.

In this work, we present Hyperbolic Hierarchical Encoding for mRNA Language Modeling (HyperHELM), a hyperbolic language-modeling framework for the CDS regions of mRNA sequences. In HyperHELM, we project token representations onto the Poincaré ball and pre-train a language model with the masked language modeling (MLM) objective directly in hyperbolic space (Figure 1). Rather than making the entire model hyperbolic, we keep the backbone Euclidean and project only

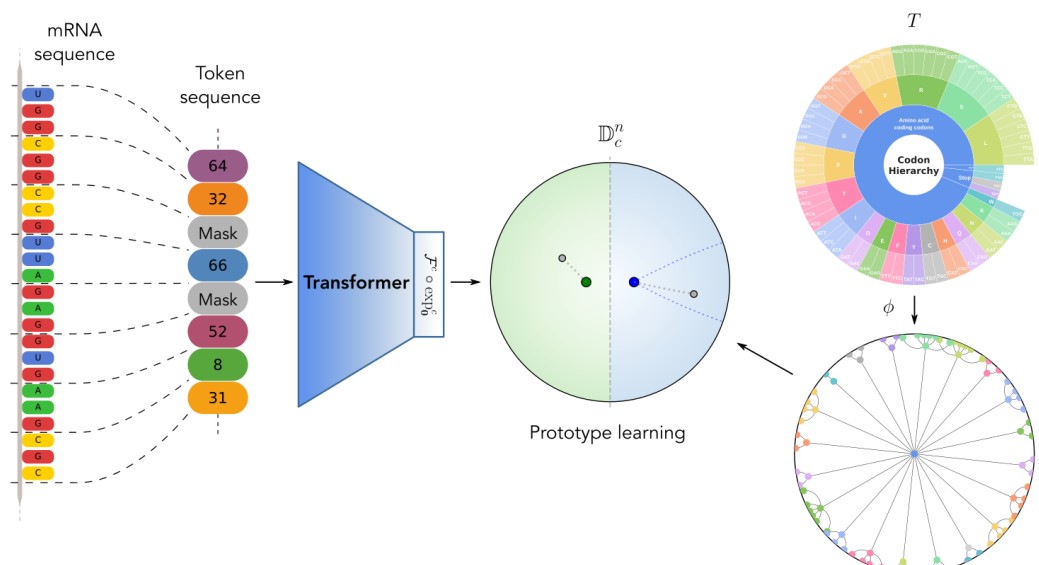

Figure 1: **High-level overview of the HyperHELM method** for MLM. The method consists of three main components: 1) the language modeling of mRNA, where a sequence transformer is used to obtain token representations, as shown in the *left*; 2) a hyperbolic embedding of the codon hierarchy (large version in Appendix B) is generated to serve as prototypes for guiding the language model during pre-training, shown on the *right*; and 3) hyperbolic hierarchical prototype learning, where the prototypes are used to predict the true label of masked tokens using either distances (*green*) or entailment cones (*blue*), visualized in the *center*.

the final-layer representations, thus retaining hardware efficiency while leveraging the hierarchical inductive bias of hyperbolic geometry.

For hyperbolic MLM pre-training, we mask a portion of input tokens and use a modular hyperbolic prediction head that scores candidates while respecting hierarchical relations. In particular, we instantiate three head options for hyperbolic learning: hyperbolic multinomial logistic regression (MLR) (Ganea et al., 2018b), distance-to-prototype learning (Snell et al., 2017), and prototype classifiers based on hyperbolic entailment cones (Ganea et al., 2018a). While Ganea et al. (2018a) primarily introduce entailment cones as a means to model hierarchical relations, our work extends this concept further by exploring its use as a similarity function instead of hyperbolic distances, aiming to capture richer relational structures. Moreover, the adaptation of these hyperbolic heads for MLM pre-training of bio-language models has never been explored before. The resulting hyperbolic latent space with hierarchy-aware MLM pre-training aligns representation geometry with the codon–amino-acid structure, clustering synonymous codons under their amino-acid parents and separating non-coding tokens (Figure 1). To our knowledge, HyperHELM is the first systematic development of hyperbolic language models for mRNA sequence data.

We conduct experiments to compare our HyperHELM with its standard Euclidean hierarchical language modeling counterparts. We keep the language model backbone architecture and pre-training dataset fixed for all models, to isolate the impact of hyperbolic geometry on hierarchy learning. We evaluate the pre-trained models on 11 diverse multi-species mRNA datasets for downstream property prediction and region annotation tasks. Across 9 out of 10 property prediction tasks, the hyperbolic approach consistently outperforms its Euclidean counterparts, even when the latter is trained to be hierarchy-aware (Yazdani-Jahromi et al., 2025a), achieving an average improvement of 10%. We also observe that in property prediction tasks, our hyperbolic language model generalizes exceptionally well to out-of-distribution data, maintaining strong performance even on long sequences with low GC-content, where standard bio-language models tend to struggle. Moreover, for the task of antibody region annotation, our HyperHELM surpasses hierarchy-aware Euclidean baseline by 3%. Our experimental results suggest that hyperbolic geometry provides a powerful inductive bias for capturing hierarchical structures in CDS regions of mRNA sequences.

To sum up, we make the following contributions:

- We explore hierarchical learning for bio-language models through the lens of hyperbolic geometry, aiming to align the structure of its representation space with the hierarchical structure of CDS regions of mRNA.

- We propose, implement, and evaluate multiple hierarchy-guided hyperbolic learning methods for masked language pre-training of a language model on CDS regions of mRNA.

- We experimentally demonstrate the benefits of hyperbolic language models on downstream mRNA property prediction and antibody region annotation, where it outperforms Euclidean models, and excels in out-of-distribution settings.

## 2 RELATED WORKS

**RNA and mRNA Models** Several supervised models for RNA and mRNA modeling exist, such as RiboNN (Zheng et al., 2025), which uses a convolutional model for predicting the translation efficiency of mature mRNA sequences; or Optimus 5-Prime (Sample et al., 2019), which is a convolutional model aimed at predicting the regulatory activity of 5' UTRs of mRNA sequences. Our focus is on unsupervised pre-training, for which the common approach is language modeling. RNA and mRNA language models enable diverse downstream tasks in property prediction, annotation, and generation. These include foundation models trained for different RNA regions such as non-coding RNA (RNA-FM (Chen et al., 2022a), RINALMO (Penić et al., 2025), and AIDO.RNA-CDS (Zou et al., 2024) which is afterwards fine-tuned to coding regions within mRNA), splice sites (SpliceBERT (Chen et al., 2023)) or UTRs (UTR-LM (Chu et al., 2024)), as well as methods using transfer learning from DNA and protein models (Prakash et al., 2024; Mollaysa et al., 2025; Garau-Luis et al., 2024) for mRNA-focused downstream tasks. For mRNA, codon-level models such as CodonBERT (Li et al., 2023) use codon tokenization with MLM to optimize coding-region embeddings. Others employ nucleotide-level tokenization, such as Orthrus (Fradkin et al., 2024), which is a Mamba-based RNA model that is pre-trained on mature RNA sequences; LoRNASH Saberi et al. (2024), which is a Hyena-based RNA model pre-trained on pre-mRNA; or Helix-mRNA (Wood et al., 2025) which employs hybrid attention and state-space architectures for improved sequence resolution and generation. Several recent models incorporate domain priors. Equi-mRNA (Yazdani-Jahromi et al., 2025b) promotes hierarchy in Euclidean space (HELM (Yazdani-Jahromi et al., 2025a)). Moskalev et al. (2024); Xu et al. (2025a;b) link sequence to structure. Despite these advances, all existing methods are confined to Euclidean spaces. To our knowledge, this is the first work to explore language model pre-training for RNA or mRNA in hyperbolic space.

**Hyperbolic learning** The exponential growth of hyperbolic space makes it a suitable domain for learning on data with an inherent hierarchical structure (Sarkar, 2011; Chamberlain et al., 2017; Nickel & Kiela, 2017). This realization has led to a surge in the popularity of hyperbolic learning (Peng et al., 2021). Deep hyperbolic architectures have been developed (Ganea et al., 2018b; Shimizu et al., 2021; Chen et al., 2022b) alongside the algorithms for optimizing such networks (Bonnabel, 2013; Bécigneul & Ganea, 2019). As a result, hyperbolic geometry has seen successful applications across many areas of machine learning, such as in computer vision (Khrulkov et al., 2020; Liu et al., 2020; Long et al., 2020; Ghadimi Atigh et al., 2021; van Spengler et al., 2023a; Mettes et al., 2024), graph learning (Liu et al., 2019; Chami et al., 2019; Zhang et al., 2021; Yang et al., 2022), Natural Language Processing (Tifrea et al., 2019; Dhingra et al., 2018) and multi-modal learning (Desai et al., 2023; Pal et al., 2025). These have shown the potential of hyperbolic learning, particularly in scenarios where the data has a clear hierarchical structure. Recently, a first work has explored the application of fully hyperbolic convolutional networks for DNA modeling Khan et al. (2025), finding that hyperbolic geometry improves genomic sequence understanding. While the structuring of mRNA is highly hierarchical in nature, existing mRNA language modeling approaches do not leverage hyperbolic geometry.

**Prototype learning** The prototype learning setting (Snell et al., 2017) has become a commonly used approach for classification tasks, where each class is represented by a prototype, resembling in some way the perfect instance of its corresponding class. Within hyperbolic learning, prototype learning approaches are mostly distinguishable by their method of obtaining prototypes (Mettes et al., 2024). Many works follow the original approach for generating prototypes based on labeled input data (Khrulkov et al., 2020; Gao et al., 2021; 2022; Guo et al., 2022). These typically create

prototypes by aggregating features of labeled instances of the corresponding class using, for example, the Fréchet mean. Another approach is to use prior knowledge of the label set to generate prototypes. Examples are (Ghadimi Atigh et al., 2021) and (Long et al., 2020), which create prototypes using a known hierarchy over the labels, or (Yu et al., 2022), which optimizes prototypes concurrently with their model through the use of known hierarchical relations. Concurrent work by (Fonio et al., 2025) generates prototypes using maximal separation, not making use of any known hierarchies. While each of these works deals with an image classification setting, we instead focus on masked language modeling. Moreover, unlike our work, none of these works explore the use of recent low-distortion embedding methods for generating prototypes from hierarchies. Lastly, except for the concurrent work by (Fonio et al., 2025), these works restrict the use of similarity functions to hyperbolic distances.

## 3  BACKGROUND ON HYPERBOLIC SPACE

In this paper we make use of the $n$-dimensional Poincaré ball model $(\mathbb{D}_c^n, \mathfrak{g})$ of hyperbolic space with constant negative curvature $-c$ and Riemannian metric $\mathfrak{g}_c^n$, where

$$\mathbb{D}_c^n = \left\{ \mathbf{x} \in \mathbb{R}^n : ||\mathbf{x}||^2 < \frac{1}{c} \right\}, \quad \mathfrak{g}_c^n = \lambda_{\mathbf{x}}^c I_n, \quad \lambda_{\mathbf{x}}^c = \frac{2}{1 - c||\mathbf{x}||^2}, \tag{1}$$

with $I_n$ being the $n$-dimensional identity matrix. For an extensive background on other isometric models and on hyperbolic geometry in general, we refer the reader to (Cannon et al., 1997; Anderson, 2006). Here, we introduce the operations that are used throughout the paper.

Using the Riemannian metric, one can compute the distances between any two points $\mathbf{x}, \mathbf{y} \in \mathbb{D}_c^n$ as

$$d_{\mathbb{D}}^c(\mathbf{x}, \mathbf{y}) = \frac{1}{\sqrt{c}} \cosh^{-1} \left( 1 + 2c \frac{||\mathbf{x} - \mathbf{y}||^2}{(1 - c||\mathbf{x}||^2)(1 - c||\mathbf{y}||^2)} \right). \tag{2}$$

Using the Möbius addition operation (Ungar, 2022), defined as

$$\mathbf{x} \oplus_c \mathbf{y} = \frac{(1 + 2c\langle \mathbf{x}, \mathbf{y} \rangle + c||\mathbf{y}||^2)\mathbf{x} + (1 - c||\mathbf{x}||^2)\mathbf{y}}{1 + 2c\langle \mathbf{x}, \mathbf{y} \rangle + c^2||\mathbf{x}||^2||\mathbf{y}||^2}, \tag{3}$$

we can define exponential and logarithmic maps (Ganea et al., 2018b)

$$\exp_{\mathbf{x}}^c : \mathcal{T}_{\mathbf{x}}\mathbb{D}_c^n \to \mathbb{D}_c^n, \quad \exp_{\mathbf{x}}^c(\mathbf{v}) = \mathbf{x} \oplus_c \left( \tanh \left( \frac{\sqrt{c}\lambda_{\mathbf{x}}^c||\mathbf{v}||}{2} \right) \frac{\mathbf{v}}{\sqrt{c}||\mathbf{v}||} \right), \tag{4}$$

$$\log_{\mathbf{x}}^c : \mathbb{D}_c^n \to \mathcal{T}_{\mathbf{x}}\mathbb{D}_c^n, \quad \log_{\mathbf{x}}^c(\mathbf{y}) = \frac{2}{\sqrt{c}\lambda_{\mathbf{x}}^c} \tanh^{-1} \left( \sqrt{c}|| - \mathbf{x} \oplus_c \mathbf{y}|| \right) \frac{-\mathbf{x} \oplus_c \mathbf{y}}{|| - \mathbf{x} \oplus_c \mathbf{y}||}, \tag{5}$$

which are used to map tangent vectors from the tangent space $\mathcal{T}_{\mathbf{x}}\mathbb{D}_c^n$ at $\mathbf{x}$ onto $\mathbb{D}_c^n$ and vice versa, respectively.

(Ganea et al., 2018b) have generalized multinomial logistic regression (MLR) to the Poincaré ball model by interpreting the MLR scores as signed distances to hyperplanes. The resulting hyperbolic MLR computes scores as

$$\ell_k(\mathbf{x}) = \frac{2}{\sqrt{c}} ||\mathbf{z}_k|| \sinh^{-1} \left( \lambda_{\mathbf{x}}^c \left\langle \sqrt{c}\mathbf{x}, \frac{\mathbf{z}_k}{||\mathbf{z}_k||} \right\rangle \cosh(2\sqrt{c}r_k) - (\lambda_{\mathbf{x}}^c - 1) \sinh(2\sqrt{c}r_k) \right), \tag{6}$$

where $\mathbf{z}_k$ and $r_k$ are the parameters corresponding to the $k$-th class. This MLR has been further extended into a hyperbolic fully connected layer $\mathcal{F}^c : \mathbb{D}_c^n \to \mathbb{D}_c^m$ by (Shimizu et al., 2021), which is computed as

$$\mathcal{F}^c(\mathbf{x}; \mathbf{Z}, \mathbf{r}) = \frac{\mathbf{w}}{1 + \sqrt{1 + c||\mathbf{w}||^2}}, \quad \mathbf{w} = \left( \frac{1}{\sqrt{c}} \sinh \left( \sqrt{c}\ell_k(\mathbf{x}) \right) \right)_{k=1}^n, \tag{7}$$

where $\mathbf{Z}$ and $\mathbf{r}$ contain the learnable parameters.

## 4 HYPERHELM

The setting that we consider is the pre-training of a CDS region of mRNA sequence model through masked language modeling (MLM) with the goal of obtaining a strong backbone for any downstream predictive task. For our approach, we take the HELM method – a language model for the hierarchical modeling of mRNA that operates fully in Euclidean space – (Yazdani-Jahromi et al., 2025a) as a starting point and replace the classifier to help guide the backbone model more effectively. More specifically, we replace the Euclidean multinomial logistic regression classifier by a hyperbolic prototypical classifier, inspired by works such as (Snell et al., 2017; Yu et al., 2022). The prototypes are generated directly from the codon-amino acid hierarchy which is shown in Figure 1 and, more clearly, in Figure 4 in Appendix B. A high-level overview of our method is given in Figure 1. Each individual component will be discussed in detail in the following subsections.

### 4.1 LANGUAGE MODELING OF MRNA SEQUENCES

Our goal is to train some sequence transformer model $f$ of CDS regions of mRNA sequences through MLM. Following recent works (Li et al., 2023; Yazdani-Jahromi et al., 2025a;b), we first apply codon-level tokenization to the mRNA sequences, where each triplet of nucleotides is represented as a single token, giving $4^3 = 64$ potential tokens, excluding special tokens. During MLM, we mask 15% of the tokens in sequences and feed these into model $f$, which outputs a representation in $\mathbb{R}^n$ for each individual token. Then, we use a classifier $g : \mathbb{R}^n \to [64]$ to predict the true label of the masked tokens. Following the HELM approach (Yazdani-Jahromi et al., 2025a), the hierarchical cross-entropy loss (Bertinetto et al., 2020) with respect to the codon hierarchy shown in Figure 1 is computed and used to update $f$ and $g$.

### 4.2 HYPERBOLIC EMBEDDINGS OF HIERARCHIES

The manner in which mRNA encodes for proteins can be understood through a hierarchy defined over the codons, visualized in Figure 1. Yazdani-Jahromi et al. (2025a) softly enforce this hierarchy in their model in Euclidean space by using the hierarchical cross-entropy loss. Here, we explicitly structure our token representation space by directly embedding the hierarchy. A hierarchy typically consists of a tree $T = (V, E)$, where the nodes $V$ contain the relevant concepts and the edges $E$ the relations between these. Moreover, we denote the leaf nodes of the tree by $L$. The tree metric $d_T$, resulting from $T$, defined as the length of the path between 2 nodes, contains the information of how strongly related any pair of concepts is. Therefore, the goal of embedding some hierarchy into a continuous space is to keep this tree metric intact. More formally, we want an embedding $\phi : V \to M$ into some connected Riemannian manifold $M$ such that $\phi$ is approximately an isometry onto $\phi(V)$, i.e.,

$$d_M\big(\phi(u), \phi(v)\big) \approx d_T(u, v). \tag{8}$$

The amount by which the metric is changed by the embedding is called the distortion. It can be shown that Euclidean spaces are unsuitable as targets for embedding trees (Sarkar, 2011), generally leading to highly distorted embeddings. Therefore, we opt to use hyperbolic space instead.

Several methods exist for embedding graphs or trees into hyperbolic space (Sarkar, 2011; Nickel & Kiela, 2017; Sala et al., 2018; van Spengler & Mettes, 2025). We embed the codon hierarchy using the HS-DTE method (van Spengler & Mettes, 2025), as it achieves the lowest distortion and thus most effectively preserves the underlying hierarchical structure, while also being very fast. Empirically, we find that the model is quite insensitive to the choice of tree embedding method (see Appendix H). We use the embeddings of the leaf nodes obtained with HS-DTE, corresponding to individual codons, as prototypes within the classifier $g$. A 2-dimensional example embedding of the entire codon hierarchy obtained with HS-DTE is shown in Figure 1.

### 4.3 PROTOTYPE LEARNING IN HYPERBOLIC SPACE

From the hierarchy embedding, we have a set of prototypes $\phi(L) \subset \mathbb{D}_c^{n_p}$ where each prototype corresponds to a particular codon and where $n_p$ is the prototype dimension. Since the embedding $\phi$ respects the tree metric $d_T$, these prototypes structure the space according to the hierarchy, without having seen any sequence data. We want to define a classifier that uses these prototypes to generate

token-level predictions. Since our backbone model $f$ outputs representations in $\mathbb{R}^n$, these are first projected onto $\mathbb{D}_c^{n_p}$ through two steps: 1) the representations are projected into hyperbolic space $\mathbb{D}_c^n$ and 2) a hyperbolic linear layer is used to project to $\mathbb{D}_c^{n_p}$. Following the convention in hyperbolic learning (Mettes et al., 2024), the first step is performed by treating the representations as tangent vectors at the origin and applying the corresponding exponential map. The second step is performed using the hyperbolic linear layer $\mathcal{F}^c : \mathbb{D}_c^n \to \mathbb{D}_c^{n_p}$ from equation 7. So, the projection can be written as

$$\mathbf{z}_i = \mathcal{F}^c\big(\exp_{\mathbf{0}}^c(\mathbf{h}_i)\big), \quad \mathbf{h}_i = f(\mathbf{t}^*)_i, \tag{9}$$

where $\mathbf{t}^*$ is the masked token sequence.

Generally, to generate token-level predictions using prototypes, softmaxed pairwise similarities between representations and prototypes are computed (Snell et al., 2017):

$$p(t_i = u|\mathbf{t}^*) = \frac{\exp\big(\beta \cdot s(\mathbf{z}_i, \phi(u))\big)}{\sum_{v \in L} \exp\big(\beta \cdot s(\mathbf{z}_i, \phi(v))\big)}, \tag{10}$$

where $\beta > 0$ is a temperature hyperparameter (set to 1.0), $t_i$ is the true $i$-th token and where $s : \mathbb{D}_c^{n_p} \times \mathbb{D}_c^{n_p} \to \mathbb{R}$ is some similarity function. Typically, negative distances $s = -d_{\mathbb{D}}$ are used as similarities, which leads the model to simply assign a token to its closest prototype. This approach is shown in Figure 2 *left*.

Alternatively, we can compute similarities using the hyperbolic entailment cone energy (Ganea et al., 2018a). Entailment cones are a geometric approach to defining hierarchical relationships in hyperbolic space. These are defined for any point $\mathbf{z} \in \mathbb{D}_c^{n_p}$ as the hyperbolic cone with $\mathbf{z}$ as its apex and with the axis of symmetry being the Euclidean straight line segment from $\mathbf{z}$ perpendicular onto the boundary of the manifold. The half aperture of the cone is

$$\psi(\mathbf{z}) = \sin^{-1}\left(\frac{K(1 - c||\mathbf{z}||^2)}{\sqrt{c}||\mathbf{z}||}\right), \tag{11}$$

where $K$ is a hyperparameter which we set to $K = 0.1$. The hyperbolic entailment cone energy is then computed as

$$E(\mathbf{x}, \mathbf{y}) = \max(0, \Xi(\mathbf{x}, \mathbf{y}) - \eta\psi(\mathbf{x})), \tag{12}$$

where $\eta > 0$ is a threshold hyperparameter (Pal et al., 2025) (set to 1.05) and where

$$\Xi(\mathbf{x}, \mathbf{y}) = \cos^{-1}\left(\frac{\langle\mathbf{x}, \mathbf{y}\rangle(1 + c||\mathbf{x}||^2) - ||\mathbf{x}||^2(1 + c||\mathbf{y}||^2)}{||\mathbf{x}|| \cdot ||\mathbf{x} - \mathbf{y}||\sqrt{1 + c^2||\mathbf{x}||^2||\mathbf{y}||^2 - 2c\langle x, y\rangle}}\right), \tag{13}$$

is the aperture required for $\mathbf{y}$ to be within the entailment cone at $\mathbf{x}$. In other words, the hyperbolic entailment cone energy is the angle by which $\mathbf{y}$ is removed from $\mathbf{x}$'s entailment cone. Examples of entailment cones and a visualization of the entailment cone energy are shown in Figure 2 *right*. The hyperbolic entailment cone energy has recently grown in popularity in areas such as vision-language learning (Desai et al., 2023; Pal et al., 2025) for encoding hierarchical relations. We propose to use both distance-based prototypes and energy-based prototypes. For both approaches, we set the negative curvature to $c = 1.0$. We also present a sensitivity analysis for the key hyperparameters in Appendix I. Lastly, we experiment using both fixed and learnable prototypes, where the prototypes are considered learnable parameters of the model, which allows learning further data-driven refinements of the hierarchy embedding. For the optimization of the learnable hyperbolic prototypes we use Riemannian SGD Bonnabel (2013). Further details regarding the optimization can be found in Appendix C.

## 5 EXPERIMENTS

In our experiments, we follow the pre-training guidelines established in HELM (Yazdani-Jahromi et al., 2025a), adopting codon-level tokenization and the masked language modeling (MLM) objective. We use the same curated OAS pre-training corpus (Olsen et al., 2022), codon vocabulary, and standard transformer backbone released in their official HELM repository [1], ensuring full comparability (detailed in Appendix C). Note that all sequences in the pre-training corpus are CDS-only sequences, with an identified open reading frame. The key difference lies in the MLM head

---

[1]https://github.com/johnsonandjohnson/HELM

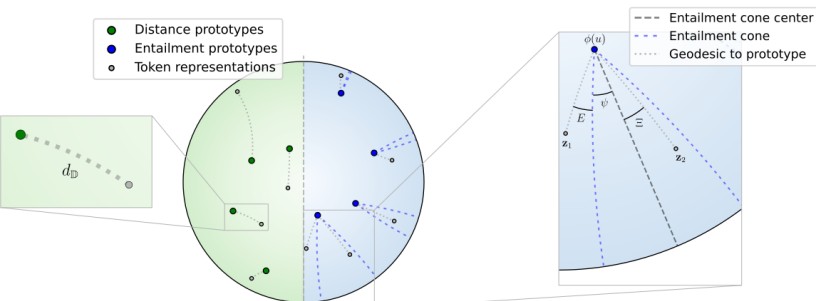

Figure 2: **Hyperbolic prototype learning.** The *center* part presents a Poincaré disk where either distances (green) or entailment cone energies (blue) are used to predict the label of embedded tokens. On the *left*, a close up of a masked token representation with its closest prototype, together with the geodesic between these is shown. The *right* part takes a closer look at one of the entailment cones, showing the geometric interpretation of equations 11, 12 and 13.

where we evaluate three hyperbolic variants: hyperbolic multinomial logistic regression, hyperbolic distance-based prototypes, and hyperbolic prototypes based on entailment cones discussed in Sections 3 and 4. We keep the rest of the method unchanged, allowing us to isolate the effect of learning the hierarchy in hyperbolic space for mRNA. For downstream tasks, we freeze the pre-trained backbone and probe the learned representations by training a TextCNN head (Kim, 2014), following standard practice (Harmalkar et al., 2023; Li et al., 2023; Yazdani-Jahromi et al., 2025a; Mollaysa et al., 2025; Yazdani-Jahromi et al., 2025b). Further experimental details are in Appendices C and E. Note that, since we only change the head of the model, the overall complexity is dominated by the backbone for each method. As a result, the difference in runtimes of the different methods is negligible (Appendix D).

**Datasets and evaluation metrics**    We use 10 datasets spanning diverse organisms and label types: Ab1 (662 antibody-encoding mRNAs) and Ab2 (2,672 antibody-encoding mRNA sequences) both with protein expression labels from Prakash et al. (2024); mRFP (1,459 sequences with protein production levels) (Nieuwkoop et al., 2023); COVID-19 Vaccine (2,400 degradation-labeled sequences) Wayment-Steele et al. (2022); *Drosophila melanogaster* (10,338 mRNA sequences) and *Saccharomyces cerevisiae* (4,937 mRNA sequences) with protein abundance labels, and *Pichia pastoris* (4,682 mRNA sequences) with transcript abundance from Outeiral & Deane (2024); Fungal (7,056 genome-derived sequences with expression labels) (Wint et al., 2022); *E. coli* (6,348 mR-NAs labeled with low/medium/high protein expression) (Ding et al., 2022); and iCodon (65,357 sequences with thermostability profiles from humans, mice, frogs, and fish) (Diez et al., 2022). Except for the *E. coli* classification task, all datasets provide regression labels for evaluating property prediction. Following prior works (Yazdani-Jahromi et al., 2025a; Li et al., 2023; Yazdani-Jahromi et al., 2025b), we use predefined train/val/test data splits and report Spearman rank correlation for regression and accuracy for classification tasks. Note that mRNA sequences in all of the downstream datasets are CDS-only, and hence contain valid codons. In general, for the sequences without identified Open Reading Frame (ORF), standard ORF-identification tools can be used (Hyatt et al., 2010; Singh & Wurtele, 2021; O'Leary et al., 2016).

**Baselines**    We evaluated HyperHELM against multiple baselines, including non-hierarchical models (Transformer XE (Yazdani-Jahromi et al., 2025b;a), Helix-mRNA (Wood et al., 2025), mRNA-FM (Chen et al., 2022a), AIDO.RNA-CDS (Zou et al., 2024) and CodonBERT (Li et al., 2023)) and the state-of-the-art, hierarchy-aware Euclidean HELM (Yazdani-Jahromi et al., 2025a). Results for additional baselines are shown and discussed in Appendix G. To ensure a fair comparison, our HyperHELM, HELM, and Transformer XE models share the same 50M-parameter backbone architecture, pre-training data, and tokenization strategy. Consequently, any observed performance differences among these models can be attributed solely to the impact of hyperbolic prototype learning.

## 5.1 HyperHELM improves downstream mRNA property prediction performance over Euclidean models

Table 1 summarizes the performance of HyperHELM variants across 10 mRNA property prediction datasets. Of these, the four HyperHELM variants achieve the best performance on 8 out of 10 datasets and the best or second best performance on 9 out of 10 datasets, demonstrating the benefits of modeling hierarchical relationships in hyperbolic spaces for mRNA sequences. Compared to the non-hierarchical Transformer XE baseline, HyperHELM improves downstream performance by 2.8–35.6%, with the largest gains observed for D. melanogaster (35.5%) and S. cerevisiae (35.6%). When compared to HELM, performance improvements range up to 32%, with particularly strong improvements on D. melanogaster (32.0%) and S. cerevisiae (20.6%) datasets. Interestingly, simple hyperbolic MLR (HyperHELM MLR) only performs well on the S.cerevisiae dataset while underperforming on all other tasks even relative to the Euclidean baselines, indicating that the combination of hyperbolic geometry with prototype-based heads is crucial for capturing hierarchical structure in mRNA embeddings. Lastly, learnable prototypes yield the best performance in 5 out of 10 datasets and either the best or second best performance in 9 out of 10 datasets, which shows that the model benefits from the freedom to refine the hierarchical embeddings during pretraining. More details on the performance of hyperbolic MLR can be found in Appendix F.

Table 1: Accuracy (for *E.coli*) and Spearman rank correlation (for all other datasets). Bold indicates the best performing model per dataset and underline indicates second best model. The missing values indicate models unable to process datasets due to sequence length limitations or OOM issues because of models being too large.

| Dataset | Non-hierarchical FMs | | | | | | Hierarchical Euclidean | Hierarchical hyperbolic (Ours) | | | | |
|---|---|---|---|---|---|---|---|---|---|---|---|---|
| | Transformer XE | Helix-mRNA | mRNA-FM | AIDO.RNA-CDS | CodonBERT | EVO-2 | HELM | MLR | Proto Dist. | Proto Entail. | Proto Dist. Learnable | Proto Entail. Learnable |
| Ab1 | 0.701 | 0.535 | 0.656 | 0.663 | 0.686 | 0.141 | 0.714 | 0.650 | 0.713 | **0.751** | 0.743 | 0.736 |
| Ab2 | 0.507 | 0.283 | 0.373 | 0.398 | 0.557 | 0.129 | 0.548 | 0.532 | 0.575 | 0.569 | **0.603** | 0.589 |
| mRFP | 0.825 | 0.432 | 0.739 | 0.787 | 0.770 | 0.239 | **0.848** | 0.744 | 0.819 | 0.802 | 0.800 | 0.820 |
| COVID-19 | 0.757 | 0.643 | 0.762 | 0.804 | 0.780 | 0.386 | 0.775 | 0.411 | 0.785 | 0.807 | **0.822** | 0.822 |
| D. melanogaster | 0.332 | - | - | - | - | - | 0.341 | 0.374 | 0.394 | **0.450** | 0.442 | 0.447 |
| S. cerevisiae | 0.354 | - | - | - | - | - | 0.398 | 0.465 | 0.434 | 0.397 | 0.424 | **0.480** |
| P. pastoris | 0.596 | - | - | - | - | - | 0.620 | 0.605 | **0.676** | 0.671 | 0.672 | 0.672 |
| Fungal | 0.690 | 0.689 | 0.722 | - | - | 0.400 | 0.702 | 0.712 | 0.735 | 0.741 | 0.742 | **0.754** |
| E. coli | 44.7 | 40.0 | **53.3** | - | - | 40.0 | 45.8 | 40.0 | 50.8 | 48.4 | 53.0 | 50.9 |
| iCodon | 0.503 | 0.157 | 0.520 | - | - | - | 0.525 | 0.517 | 0.535 | 0.539 | **0.545** | 0.536 |

## 5.2 Codon usage Bias/Pattern is an indicator for hyperbolic model gains

We observed that HyperHELM's performance gains vary significantly across datasets (Table 1). Building on prior work that links gains from hierarchical learning to codon usage bias (Yazdani-Jahromi et al., 2025a), we investigated if this holds for models trained in hyperbolic spaces.

To this end, we measured each dataset's synonymous codon usage bias using the Effective Number of Codons (ENC) metric (Wright, 1990). This metric quantifies codon diversity: a low ENC value signifies high bias (a strong preference for specific codons for a given amino acid), while a high value indicates codons are used more uniformly. As shown in Figure 3, our results confirm the hypothesis: datasets with greater codon usage bias (lower ENC) consistently achieve larger gains with both HyperHELM prototype based variants. Intuitively, this is because a strong codon bias creates a stronger learnable hierarchical pattern even among synonymous codons beyond the hierarchy defined by codons and amino acids. This additional hierarchy is naturally suited to the geometry of hyperbolic space, allowing HyperHELM to capture these dependencies from data more effectively than non-hierarchical models.

## 5.3 HyperHELM improves Antibody Sequence Annotation

We further assess HyperHELM on the task of antibody (Ab) sequence region annotation, a benchmark introduced in prior work (Yazdani-Jahromi et al., 2025a), important for immunological studies (Briney & Burton, 2018). This task involves predicting the identity of nucleotides in Ab-coding mRNA into one of four biologically meaningful regions: signal peptides, V, DJ, or constant regions.

We use the same held-out test set of 2000 curated antibody sequences as used in Yazdani-Jahromi et al. (2025a) for this task and compare our prototype based HyperHELM models against the HELM baseline. As shown in Table 2(a), both HyperHELM variants outperform Euclidean HELM, with the prototype distance model achieving the best accuracy of 76.48%, and the prototype entailment variant being second best with accuracy of 75.21%, compared to 73.48% achieved by HELM. The

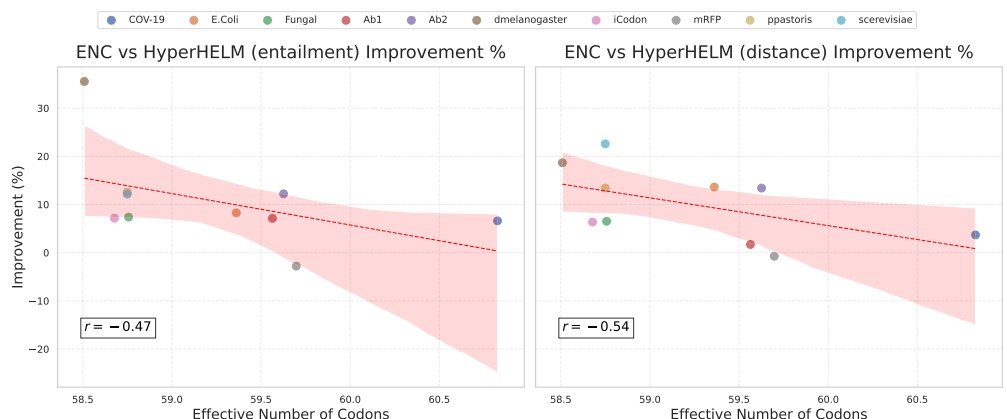

Figure 3: Relationship between codon usage metric (ENC) and HyperHELM performance gains. Hyperbolic gains are largest for sequences with higher codon usage bias indicated by lower ENC.

results highlight the advantage of hierarchy-aware learning in hyperbolic space to effectively capture the structure of antibody mRNA regions.

## 5.4 Impact of Sequence Length and GC Content on Model Performance

We examine model robustness across different biologically meaningful mRNA sequence characteristics by stratifying datasets according to sequence length and GC content. These factors are known to be relevant for mRNA engineering (Courel et al., 2019; Zhang et al., 2011; Jia & Qian, 2021) and have been linked to differences in model generalization (Castillo-Hair & Seelig, 2021; Qiu, 2023; Szikszai et al., 2022). Longer sequences often contain more complex dependencies and are underrepresented in training data, while extreme GC content alters secondary structure; both scenarios making it challenging for models to learn effectively.

**Sequence Length Analysis**  We analyzed performance on the *Pichia pastoris* dataset by dividing sequences into three length categories: *short* (30–1000 nucleotides), *medium* (1000–2000 nucleotides), and *long* (2000–3000 nucleotides). Since the pre-training data consists of sequences around 1400 nucleotides (a typical range for mRNA vaccines (Gunter et al., 2023)), the long sequences represent an out-of-distribution (OOD) challenge.

As shown in Table 2(b), Euclidean HELM's performance degrades sharply with increasing length, consistent with prior findings (Yazdani-Jahromi et al., 2025a). In contrast, both HyperHELM variants reverse this trend, with performance improving on long sequences compared to medium ones. The entailment-based variant reached a Spearman correlation of 0.70 (a +0.24 absolute improvement over HELM), while the distance-based variant showed a +0.19 improvement. This indicates that HyperHELM's hyperbolic-space representation is beneficial even for out-of-distribution length shifts, a trend also reported for hyperbolic models in other domains (Ibrahimi et al., 2024; Kasarla et al., 2025).

**GC Content Analysis**  For the COVID-19 dataset, we categorize sequences based on GC content into: *low* (GC $\leq 47\%$), *medium* ($47\% < $ GC $\leq 55\%$), and *high* (GC $> 55\%$). These thresholds align with widely used biological definitions, where GC content below $47\%$ is considered low and above $55\%$ is high (Brown, 2007; Courel et al., 2019).

Performance for both HELM and HyperHELM (shown in Table 2(c)) is reasonably high in the low GC range but diminishes for high GC content sequences due to their relative scarcity in the pre-training corpora. Notably, the entailment-based HyperHELM attains a Spearman rank correlation of 0.62 in the high GC category compared to HELM's 0.56, and achieves a strong Spearman rank correlation of 0.73 in the medium GC category, a gain of +0.09 over HELM.

Table 2: (a) Accuracy of antibody sequence region annotation, (b) Spearman rank correlation across sequence lengths for *P. pastoris*, (c) Spearman rank correlation across GC content for the COVID-19 dataset. Best performance is shown in bold.

| Model | Acc. (%) |
|---|---|
| HELM | 73.48 |
| HyperHELM (Dist.) | **76.48** |
| HyperHELM (Entail.) | 75.21 |

(a) Antibody annotation

| Model | Short | Med. | Long |
|---|---|---|---|
| HELM | 0.54 | 0.58 | 0.46 |
| HyperHELM (Dist.) | **0.65** | **0.59** | 0.65 |
| HyperHELM (Entail.) | 0.61 | 0.56 | **0.70** |

(b) Sequence length analysis

| Model | Low | Med. | High |
|---|---|---|---|
| HELM | **0.78** | 0.64 | 0.56 |
| HyperHELM (Dist.) | 0.77 | 0.62 | 0.54 |
| HyperHELM (Entail.) | **0.78** | **0.73** | **0.62** |

(c) GC content analysis

## 6 CONCLUSION

The strong performance of our hyperbolic prototype based models indicates that explicitly modeling hierarchical mRNA relationships in hyperbolic space is more effective than standard Euclidean approaches, even when the latter are made hierarchy-aware. Hyperbolic embeddings not only improve downstream property prediction but also offer a more faithful reflection of codon-amino-acid relationships, particularly in sequences with strong codon usage bias. Results also demonstrate that hyperbolic hierarchy-aware modeling can help generalization to out-of-distribution settings such as modeling long sequence lengths and low GC contents. The observed improvements highlight the potential of hybrid language models for biological sequences, where Euclidean backbones are paired with hyperbolic heads, as a practical strategy to integrate hierarchical inductive biases without incurring the computational overhead of fully hyperbolic networks.

**Limitations and Future Work** Our current HyperHELM variants use fixed prototypes; future work will explore making these prototypes learnable during training. We also plan to extend our methods to Causal Language Modeling for generative applications. Other promising directions include applying hyperbolic models to different biological modalities, such as protein and genomic sequences, and investigating adaptive or mixed-geometry latent spaces.

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

# A STATEMENT OF REPRODUCIBILITY

All datasets evaluated in this work were introduced in previously published studies, which are properly cited in the main text. Upon acceptance of the paper, we will release the codebase required to reproduce the experiments, including data preprocessing scripts, training and evaluation pipelines, and configuration files, along with the pretrained model weights. This will ensure full transparency and enable exact reproduction of our results.

# B HIERARCHICAL RELATIONSHIP OF CODONS AND AMINO ACIDS IN MRNA

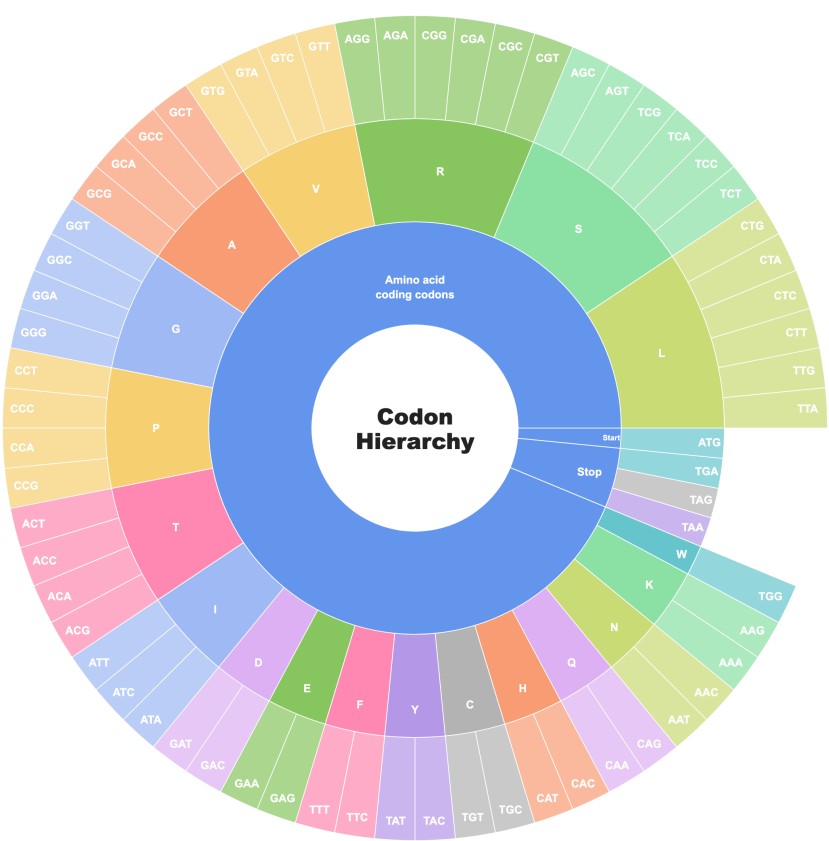

Figure 4: The codon hierarchy that is used for creating prototypes and structuring the representation space.

# C PRE-TRAINING DETAILS

**Model and training**    All our experiments were run with a transformer backbone, consisting of 10 transformer layers with an intermediate size of 2560 and a hidden size of 640, resulting in a total of ∼50M parameters. All models were pretrained for 40 epochs with a batch size of 1024 spread across 8 Nvidia A100 GPUs using the hierarchical cross-entropy (HXE) loss with respect to the codon hierarchy shown in Figure 4 following (Yazdani-Jahromi et al., 2025a).

Sequences were tokenized using codon-level tokenization, resulting in vocabulary size of 70, including special tokens. The maximum context-length was set to 444, which is enough to accommodate

all sequences in the pretraining dataset. However, the positional embedding layer was configured to support up to 2048 tokens, as such longer sequences can appear in certain downstream tasks. Positional embedding was applied following the strategy from GPT-2 (Radford et al., 2019).

Optimization was performed using the AdamW optimizer (Loshchilov & Hutter, 2019) with a weight decay of 1e-1. The learning rate was scheduled using linear warmup, followed by cosine decay, using an initial learning rate of 1e-4 which decayed to a minimum of 1e-5. Following (Yazdani-Jahromi et al., 2025a), the $\alpha$ of the HXE loss was set to 0.2.

For the prototype classifiers, we used a prototype embedding dimension of 128 and used a scaling factor $\tau = 2.0$ for the embedding with h-MDS (van Spengler & Mettes, 2025). A hyperbolic linear layer (Shimizu et al., 2021) was used to project to the representation space. The temperature $\beta$ was set to 10. When the prototypes are made learnable, their optimization is performed using Riemannian SGD Bonnabel (2013), which performs updates as

$$\mathbf{p}_i^{(t+1)} = \exp_{\mathbf{p}_i^{(t)}}^c (\alpha \nabla_{\mathbf{p}_i^{(t)}} \mathcal{L}), \tag{14}$$

where $\mathbf{p}_i^{(t)}$ is the $i$-th prototype at $t$ iterations, where $\nabla_{\mathbf{p}_i^{(t)}} \mathcal{L}$ is the gradient of the loss evaluated at $\mathbf{p}_i^{(t)}$ and where $\alpha$ is the learning rate. The learning rate is scheduled identically to the learning rate of the AdamW optimizer. The hyperbolic operations were implemented using the HypLL library van Spengler et al. (2023b).

**Pre-training corpus** The pre-training corpus consists of the curated OAS database (Olsen et al., 2022) adopted from HELM (Yazdani-Jahromi et al., 2025a). For completeness and self-consistency, the curation procedure is summarized here.

The full OAS database contains more than two billion unpaired and around two million paired antibody sequences from various species, each with a known open reading frame. However, the raw database exhibits a high degree of sequence redundancy and includes a non-trivial fraction of functionally invalid sequences (e.g., sequences with frameshifts, truncations, or non-canonical residues). To obtain a high-quality pre-training corpus, the filtering strategy introduced in HELM is followed.

First, filtering based on the *ANARCI status* annotation provided in OAS is applied, excluding sequences with unusual residues, indels, truncations, or missing conserved cysteines, all of which are often indicative of problematic or non-functional sequences. Sequences with V and J gene identity below 0.7 are then discarded, ensuring a high degree of similarity to known reference germline genes. Only sequences labeled as *productive* and *complete vdj* are retained, indicating that the corresponding sequences are fully functional.

The corpus is subsequently restricted to human antibodies by applying a species filter. To reduce redundancy, sequence similarity clustering using Linclust (Steinegger & Söding, 2018) is performed independently on paired and unpaired sequences with a sequence identity threshold of 0.5, and only the cluster centroids are kept as representatives. Because paired sequences are much fewer in number than unpaired ones, paired antibodies are split into their heavy and light chains and further treated as unpaired. Finally, heavy chains are subsampled to approximately match the number of light chains while maintaining realistic gene frequency distributions.

This process yields a curated pre-training corpus of 15.3 million mRNA sequences, comprising 7.7 million heavy-chain and 7.6 million light-chain CDSs.

## D RUNTIME COMPARISON OF PRE-TRAINING METHODS

Table 3 shows the runtime in minutes per epoch for each of the methods on $8\times$Nvidia A100 GPUs as obtained using the pre-training setting discussed in detail in Appendix C. As expected, the runtimes of each method are rather similar, due to the identical backbones dominating the computational complexity.

Table 3: Comparison of the runtime between the different methods that were used for pre-training.

|  | Transformer XE | HELM | MLR | Proto Dist. | Proto Entail. |
|---|---|---|---|---|---|
| **Runtime** (min/epoch) | 73.2 | 71.1 | 71.7 | 72.2 | 73.1 |

## E    DOWNSTREAM TASKS DETAILS

For downstream evaluation, we used a TextCNN (Kim, 2014) for each downstream task, following (Marquet et al., 2022; Chen et al., 2024; Outeiral & Deane, 2024; Harmalkar et al., 2023; Yazdani-Jahromi et al., 2025a). Our downstream configuration exactly matches that of (Yazdani-Jahromi et al., 2025a). So, we use a hidden size of 640 and 100 channels in the convolutions. The pretrained weights of the backbone are frozen during training. For each model we perform a hyperparameter search on the grid spanned by learning rates of 3e-4, 1e-4, 1e-5 and batch sizes 8, 16, 32, 64. The optimal hyperparameter configuration was chosen based on an unseen validation set. The final reported performance is determined on a separate test set. Each downstream dataset is split into 70% training, 15% validation and 15% test data.

## F    PERFORMANCE OF HYPERBOLIC MLR

As shown in the results in Table 1, hyperbolic MLR performs poorly even when compared to the Euclidean baselines, showcasing that simply replacing the geometry by hyperbolic geometry is not sufficient for improving performance. This poor performance is likely due to the numerical problems that occur near the boundary of the space. Training using MLR causes the representations of each token to be pushed towards the hyperplane corresponding to their class and then beyond it. As a result, as training progresses, the representations obtained by MLR rapidly grow in norm, causing these to end up in the region of numerical instability. HypLL van Spengler et al. (2023b) and other hyperbolic libraries deal with this potential numerical problem by clipping points to a region within which numerical issues will certainly not arise. However, this means that when training with MLR, representations often get clipped after a few iterations, destroying all the information stored in the norms. This effect can be seen in Figure 5. We suspect that this severely hinders further learning past this point, resulting in poor performance.

Note that both our proposed prototype methods do not suffer from this issue, as observed for the entailment method in Figure 5. For distance prototypes this is quite straightforward, since token representations are pushed towards their corresponding prototype, which itself has a relatively small norm. For entailment cones, once a token representation lies within the cone of its corresponding prototype, the embedding is no longer pushed away from the origin.

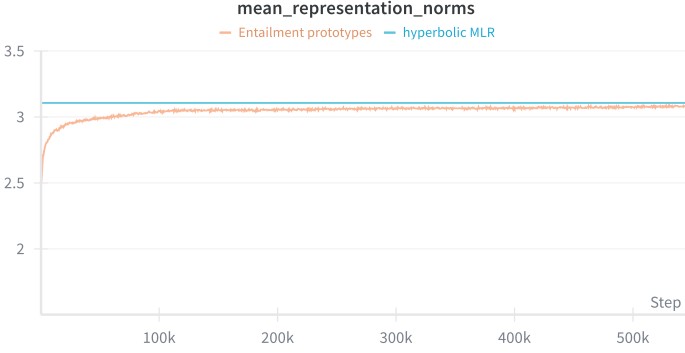

Figure 5: The mean hyperbolic norm of representations at each training step during pretraining using prototypes with entailment cones or hyperbolic MLR.

## G ADDITIONAL BASELINES

Table 4 contains results for the downstream prediction tasks with additional baselines: linear regression with 1-hot embeddings at codon-level, Euclidean and hyperbolic CNNs with codon-level tokenization, RNA-FM (Chen et al., 2022a), SpliceBERT (Chen et al., 2023) and EVO 2 (7B) (Brixi et al., 2025). Because of the large sizes of the foundation models, not all experiments were feasible due to hardware constraints. Moreover, SpliceBERT cannot be applied to several datasets due to the maximal sequence length that it can handle. Our method performs best in all cases. Note that the foundation model baselines were pretrained on different corpora, making the comparison less relevant than the comparison shown in Table 1. The particularly poor performance of EVO 2 (7B) can be explained by it being a general model not specialized to mRNA.

Table 4: Accuracy (for *E.coli*) and Spearman rank correlation (for all other datasets) for additional baselines. Bold indicates the best performing model per dataset and underline indicates second best model. The missing values indicate inability to perform experiment due to hardware constraints.

| Dataset | Linear regression | Euclidean CNN | Hyperbolic CNN | RNA-FM | SpliceBERT | EVO 2 (7B) | Proto Dist. | Proto Entail. |
|---|---|---|---|---|---|---|---|---|
| Ab1 | 0.582 | 0.421 | 0.518 | 0.595 | 0.652 | 0.129 | 0.713 | **0.751** |
| Ab2 | 0.499 | 0.243 | 0.252 | 0.515 | 0.542 | 0.141 | **0.575** | 0.569 |
| mRFP | 0.687 | 0.474 | 0.193 | 0.527 | 0.596 | 0.239 | **0.819** | 0.802 |
| COVID-19 | 0.545 | 0.602 | 0.480 | 0.742 | 0.757 | 0.386 | 0.785 | **0.807** |
| D. melanogaster | 0.123 | 0.103 | 0.037 | - | - | - | 0.394 | **0.450** |
| S. cerevisiae | 0.285 | 0.143 | 0.121 | - | - | - | **0.434** | 0.397 |
| P. pastoris | 0.553 | 0.301 | 0.229 | - | - | - | **0.676** | 0.671 |
| Fungal | 0.475 | 0.606 | 0.580 | - | - | 0.400 | 0.735 | **0.741** |
| *E. coli* | 37.7 | 40.0 | 40.0 | - | - | 40.0 | **50.8** | 48.4 |
| iCodon | 0.391 | 0.152 | 0.143 | - | - | - | 0.535 | **0.539** |

## H SENSITIVITY ANALYSIS WITH RESPECT TO THE HYPERBOLIC TREE EMBEDDING METHOD

Table 5 shows results on several downstream datasets obtained when using fixed entailment prototypes generated using either Poincaré embeddings Nickel & Kiela (2017) or HS-DTE van Spengler & Mettes (2025). As can be seen, both approaches result in similar performance, showcasing that our method is insensitive to the quality of the embdding method.

Table 5: Spearman rank correlation for several datasets obtained using fixed entailment prototypes generated using either Poincaré embeddings or HS-DTE.

| Dataset | Poincaré Embeddings | HS-DTE |
|---|---|---|
| Ab1 | 0.752 | 0.751 |
| Ab2 | 0.546 | 0.569 |
| mRFP | 0.829 | 0.802 |
| COVID-19 | 0.820 | 0.807 |
| Fungal | 0.728 | 0.741 |

## I SENSITIVITY ANALYSIS WITH RESPECT TO CHOICE OF HYPERPARAMETERS

To evaluate the robustness of our hyperbolic modeling approach, we performed a sensitivity analysis examining variations in curvature and threshold hyperparameters. The results, summarized in Table 6, indicate that the model's performance is relatively stable across the tested ranges.

Across most datasets, changes in hyperparameters lead to minor fluctuations in performance, demonstrating that the model does not rely heavily on precise hyperparameter tuning within this scope. For example, the performance on COVID-19, Ab1, and Fungal, the performance varies by a few percentage points across different hyperparameter settings.

Table 6: Sensitivity of model performance to hyperparameter variations.

| Dataset | $c$=0.20, $\eta$=1.05 | $c$=0.50, $\eta$=1.05 | $c$=1.00, $\eta$=1.1 | $c$=1.00, $\eta$=1.2 | $c$=1.00, $\eta$=1.05 |
|---|---|---|---|---|---|
| COVID-19 | 0.779 | 0.816 | 0.800 | 0.806 | 0.807 |
| Ab1 | 0.739 | 0.742 | 0.717 | 0.724 | 0.751 |
| Ab2 | 0.593 | 0.584 | 0.578 | 0.583 | 0.569 |
| Fungal | 0.733 | 0.748 | 0.733 | 0.732 | 0.741 |
| P. pastoris | 0.667 | 0.650 | 0.678 | 0.680 | 0.671 |

## J    RELATION TO HYPERBOLIC GENOME EMBEDDINGS

While both Hyperbolic Genome Embeddings (HGE) (Khan et al., 2025) and HyperHELM explore the use of hyperbolic geometry for biological data, the settings and goals differ in several important ways. Firstly, HGE is a DNA model while HyperHELM specifically focuses on mRNA. Secondly, HGE proposes a fully hyperbolic architecture trained directly on specific downstream tasks, whereas HyperHELM is designed as a self-supervised language model for large-scale pre-training on unlabeled mRNA sequences, with the goal of learning general-purpose representations that can be reused across many downstream tasks. Consequently, our primary conceptual and experimental comparisons are to other bio-language models that perform MLM-style pre-training, rather than to task-specific supervised architectures. This focus on large-scale pre-training also drives key technical choices that differ from HGE: in particular, we find that hyperbolic prototype-based heads are crucial for MLM performance, while hyperbolic MLR alone underperforms, and we adopt a projection-based approach (Euclidean backbone with hyperbolic heads) rather than a fully hyperbolic network to maintain scalability and compatibility with standard transformer pre-training.

## K    ABLATION ON "DOUBLE-DIPPING"

We want to verify that our method is not "double-dipping" on hierarchical information by using hierarchical cross-entropy loss, and potentially introducing redundancy or conflicting optimization signals. To evaluate this, we performed an ablation comparing our method when used with (i) standard cross-entropy (XE) loss and (ii) hierarchical cross-entropy (HXE) loss. We trained the same model architecture under both loss configurations, keeping all other training conditions identical. This allows us to isolate the effect of the loss function on performance and determine whether hierarchical information is being over-used or inconsistently exploited. The results show that combining our method with standard XE leads to lower performance in general. In contrast, pairing our method with HXE yields improved performance. This confirms that HXE provides a more coherent optimization signal and does not introduce conflicting gradients with our method. In other words, the hierarchical supervision is complementary rather than redundant.

| Dataset | Proto Dist. (HXE) | Proto Dist. (XE) |
|---|---|---|
| ppastoris | **0.676** | 0.666 |
| scerevisiae | **0.434** | 0.342 |
| mRFP | **0.819** | 0.752 |
| E. Coli | **50.8** | 48.6 |
| Fungal | 0.735 | **0.740** |
| COVID-19 | **0.785** | 0.775 |

Table 7: Comparison of Proto Distance under hierarchical cross-entropy (HXE) and standard cross-entropy (XE).

## L    EFFECT OF THE BASE POINT OF THE EXPONENTIAL MAP

To examine whether centering the exponential map at the origin may introduce an information bottleneck, we conducted additional experiments in which the base point was made fully learnable. For a fair comparison, we fixed the entailment prototypes and directly compared this learnable–base model with the origin-centered mapping used in the main paper. Across nine datasets, the origin-centered model performs better in 4 out of 9 cases, is on par in 4 out of 9, and is worse in only 1 out of 9. These results indicate that learning the base point does not yield consistent improvements, and that the origin choice is not a bottleneck in practice. These findings support our choice of using the origin as the base point: it aligns naturally with the hierarchical geometry and performs as well as, or better than, a learnable alternative.

| Dataset | Origin base | Learnable Base |
|---|---|---|
| Ab1 | **0.751** | 0.701 |
| S. cerevisiae | **0.397** | 0.369 |
| COVID-19 | **0.807** | 0.783 |
| Fungal | **0.741** | 0.724 |
| E. Coli | 48.4 | **50.6** |
| D. melanogaster | **0.450** | **0.451** |
| Ab2 | **0.569** | **0.570** |
| mRFP | **0.802** | **0.805** |
| P. pastoris | **0.671** | **0.671** |

Table 8: Comparison between a learnable base point and the origin-centered exponential map.

