# OpenReview forum: "HyperHELM: Hyperbolic Hierarchy Encoding for mRNA Language Modeling"
_ICLR.cc/2026/Conference — Submitted to ICLR 2026_

### Official Review · Reviewer_5EoM · 2025-10-22

**Soundness:** 3
**Presentation:** 3
**Contribution:** 2
**Rating:** 2
**Confidence:** 5

**Summary:**

This paper extends the work presented in “HELM: Hierarchical Encoding for mRNA Language Modeling” [1] by replacing the hierarchical encoding in Euclidean space with one in hyperbolic space. Hyperbolic geometry is leveraged as an inductive bias for hierarchical language modeling of mRNA sequences. Following the architecture described in [1], the proposed HyperHELM model employs a Euclidean backbone, with only the final layer replaced by a hyperbolic prototypical classifier. This design preserves hardware efficiency while exploiting the representational advantages of hyperbolic geometry. The paper investigates three approaches to hyperbolic learning: hyperbolic multinomial logistic regression (MLR), distance-to-prototype learning, and prototype classifiers based on hyperbolic entailment cones. It is the first work to explore hyperbolic language modeling for mRNA sequence data, demonstrating clear benefits over Euclidean space-based models across multiple downstream tasks.

[1] Mehdi Yazdani-Jahromi, Mangal Prakash, Tommaso Mansi, Artem Moskalev, and Rui Liao. HELM: Hierarchical Encoding for mRNA Language Modeling. In *International Conference on Learning Representations*, 2025.

**Strengths:**

- Overall, the paper is well written, with a clear central idea and an effective flow that guides the reader through the motivation and methodology.
- The paper presents **HyperHELM**, the first mRNA language model pretrained using masked language modeling in hyperbolic space. While the concept of hierarchical encoding for mRNA sequence data was previously introduced in [1], the application of hyperbolic geometry to RNA sequence modeling is novel.
- HyperHELM is evaluated on ten diverse downstream tasks spanning multiple organisms and covering various mRNA-related properties, such as protein production, expression levels, degradation, and abundance. It achieves the best performance on nine out of ten datasets, underscoring the advantages of hierarchy-aware learning in hyperbolic space.
- Unlike the Euclidean space-based HELM, which struggles with longer RNA transcripts, HyperHELM maintains or even improves its performance as sequence length increases. This finding is consistent with prior reports of hyperbolic models in other domains and is particularly important, as longer mRNA sequences are often underrepresented in datasets and pose challenges for existing models.

**Weaknesses:**

- **Novelty:** While the idea of using hyperbolic geometry has not previously been applied to mRNA sequence modeling, it has been successfully explored across many areas of machine learning. The application of hyperbolic geometry to mRNA sequence encoding is well motivated; however, the theoretical framework used in the paper does not contribute new insights to the broader machine learning field. As such, the paper’s novelty lies primarily in its application rather than in methodological innovation.
- **Reproducibility:** The paper does not mention whether the code or pretrained model weights will be made publicly available. For application-oriented work, this should be considered essential. Moreover, the dataset splits used for downstream tasks are not provided. This omission is particularly concerning, given that labeled mRNA datasets are typically small (often fewer than 5,000 samples). Repeating experiments with different random splits can lead to significant variation in performance.
- **Downstream task results**: Table 1 reports Spearman’s rank correlation and accuracy for ten tasks, but only single numerical values are provided, without clarification on whether these represent mean results or single runs. Combined with the lack of details on dataset splits, this makes the results potentially unreliable. For such small datasets, it is standard practice to perform cross-validation or multiple runs with different splits, reporting both mean and standard deviation to ensure robustness and statistical validity.
- **Baselines**: HyperHELM is compared to five baselines on four downstream tasks, but only two baselines on the remaining six. The chosen baselines include RNA-FM, SpliceBERT, and CodonBERT. Since the authors use a simple probing setup (with frozen language model parameters), comparisons to RNA-FM, a model pretrained exclusively on non-coding RNAs, are not particularly informative. Similarly, SpliceBERT was pretrained on pre-mRNAs, making mRNA data effectively out-of-distribution. While these models may perform well when fine-tuned on new RNA types, such comparisons are not meaningful under a probing-only setup. Additionally, simpler baselines such as linear or logistic regression are missing and would help contextualize the gains of the proposed approach.
- **References**: Several references appear unrelated to the core topic (for example, *Xu et al.* in line 131). There also appears to be an overrepresentation of citations from a single research group, some of which are only tangentially related and not essential for understanding the paper. A more balanced and focused selection of references would strengthen the scholarly framing.

I recommend **rejecting** the paper.

1. The paper lacks methodological novelty in machine learning, as it mainly extends an existing framework (HELM) by replacing Euclidean space with hyperbolic space.
2. Reproducibility is limited due to the absence of publicly available code, pretrained weights, and dataset splits, raising concerns about result reliability.

While the paper is clearly written and the motivation to use hyperbolic geometry for mRNA sequence modeling is well justified, the contribution is primarily applied rather than methodological. The theoretical and algorithmic components of hyperbolic learning are already well established in prior literature, and the paper does not introduce new learning principles or modeling techniques.

Empirically, the reported results on downstream tasks are promising - HyperHELM outperforms Euclidean baselines on 9 out of 10 datasets and shows robustness to longer RNA sequences. However, the absence of code, pretrained weights, and dataset splits makes it difficult to verify or reproduce these findings. Moreover, the use of small datasets without clear information on data splits or repeated trials reduces the statistical reliability of the reported improvements.

Overall, the work is a solid application of existing hyperbolic methods to a new biological domain, but it falls short of the novelty and reproducibility standards expected for acceptance at a top-tier conference. The paper, in this form, is more suitable for presentation at a workshop than a main conference.

**Questions:**

- **Availability:** Will the code, pretrained model weights, and downstream task dataset splits be made publicly available if the paper is accepted?
- **Reproducibility:** Please comment on the reproducibility of the work. Are sufficient details provided to replicate the experiments?
- **Downstream task results:** In Table 1, please clarify what the reported numbers represent. Are these results from single runs, or do they correspond to mean values across multiple experiments or cross-validation folds?
- **Baselines**: Can you compare HyperHELM to simpler baselines? I think having proper baselines would improve the overall presentation of the results.
- **HXE loss parameter:** Have you experimented with different values of the hyperparameter $ \alpha $ in the HXE loss? It appears to be an important factor, yet it was fixed at 0.2 for all experiments. Please elaborate on the reasoning behind this choice and whether sensitivity analysis was performed.

---

> ### Author Response · Authors · 2025-11-20
> **Response to Review-Response 1/3**
>
> We thank the reviewer for their compliments regarding the writing, the motivation of our method and its efficacy. Below we address the concerns raised by the reviewer, particularly regarding the novelty and reproducibility, and answer their questions.
>
> **Novelty.**
> To our knowledge, HyperHELM is the first method to perform language modeling with hyperbolic prototypes (even beyond mRNA modeling), which turns out to be a key ingredient for robust pre-training. Moreover, our combination of prototype learning with hyperbolic entailment cones and the use of HS-DTE to generate these prototypes for a biologically grounded label hierarchy is also novel. Importantly, simply replacing the geometry of HELM by Euclidean geometry leads to the hyperbolic MLR method, which performs very poorly, as shown in Table 1 of the main text. Thus, we disagree with the reviewer’s claim that our work “mainly extends an existing framework (HELM) by replacing Euclidean space with hyperbolic space,” as our method is significantly more involved than a simple change of geometry. During the rebuttal, we have now also added learnable prototypes which further strengthen our results (in responses to reviewers bM9n and Ak6U).
>
> We also note that our paper was submitted to the "applications to physical sciences (physics, chemistry, biology, etc.)" track of ICLR 2026 (specifically mentioned in ICLR 2026 call for papers (https://iclr.cc/Conferences/2026/CallForPapers)), where the goal is precisely to bring such methodological advances into challenging scientific domains and to demonstrate concrete benefits there. In this context, the improved results we obtain clearly showcase the contribution of the proposed methodology.
>
> We hope the reviewer is willing to reconsider their perspective based on the above, and we remain open to further discussion.
>
> **Reproducibility.**
> We really appreciate the reviewer's focus on the reproducibility of scientific work. The complete code used for all of our experiments will be made available, along with pre-trained weights, after acceptance of the paper. Because of our institutional policy on code release, we are not allowed to release it yet, as is common with many institutions. We want to additionally commit to reproducibility with the explicit Statement of Reproducibility in Appendix A, confirming our intention to release the code upon acceptance.
>
> **Downstream task results.**
> In our experiments, we rely on public pre-defined dataset splits to make comparisons aligned with the published results of the existing models. Specifically, mRFP, COVID-19, Fungal, _E. coli_ and iCodon dataset splits are taken from CodonBERT's [1] GitHub release ; Ab1 and Ab2 splits come from HELM [2]; D. melanogaster, P. pastoris, and S. cerevisiae splits are adopted from CaLM [3]. We do not change seeds or splits to ensure comparability with prior works as has been done in recent works including [1,2,3,4,5].
>
> **Comparison to additional baselines.** We thank the reviewer for suggesting additional baselines. Based on their suggestions, we have included several additional baselines, namely EVO 2 (7B), Helix-mRNA, mRNA-FM, and AIDO.RNA-CDS. The results are shown in the table below. Due to the computational and time constraints, we are not able to obtain results of all models on all downstream datasets. As can be seen, our model outperforms the additional baselines, including EVO 2, on all of the datasets except for AIDO.RNA-CDS on COVID-19, where our model performs on par, and mRNA-FM on _E. coli_ where mRNA-FM is the strongest model. These additional results have been included in the paper. Missing value for Fungal dataset with AIDO.RNA-CDS indicates the runs did not execute due to hardware limitations. We have also included these in Appendix Section G.
>
>
> | Dataset       | Helix-mRNA |  mRNA-FM | EVO 2 | AIDO.RNA-CDS | Proto Dist. (Ours)     | Proto Entail. (Ours)|
> |---------------|:----------:|:--------:|:-----:|:------------:|:----------------------:|:------------------------:|
> | Ab1           |    0.535   |   0.656  | 0.129 |     0.663    |    0.713               |   **0.751**              |
> | Ab2           |    0.283   |   0.373  | 0.141 |     0.398    |  **0.575**             |     0.569                |
> | mRFP          |    0.432   |   0.739  | 0.239 |     0.787    |  **0.819**             |     0.802                |
> | COVID-19          |    0.643   |   0.762  | 0.386 |     0.804    |    0.785               |   **0.807**              |
> | Fungal        |    0.689   |   0.722  | 0.400 |              |    0.735               |   **0.741**              |
> | Ecoli         |    40.0    | **53.3** |  40.0 |       52.9       |     50.8               |      48.4                |

---

> ### Author Response · Authors · 2025-11-20
> **Response to Review-Response 2/3**
>
> **Choice of HXE loss parameter $\alpha$.**
> The choice for the $\alpha = 0.2$ parameter was based on the ablation results found in HELM [2]. Most importantly, they report in their Appendix that a value of 0.2 performs best and that the results on most datasets are not too sensitive to this choice of $\alpha$. We simply adopted it from them for fair comparison.
>
> **References.**
>
> Following the suggestions of the reviewers we have added references for more baselines in the related work section of the paper. If the reviewer knows of any other paricular references that should have been added, then we would be more than happy to add these as well.
>
> [1] Li, Sizhen, et al. "CodonBERT: Large Language Models for mRNA design and optimization." Genome Research, 2023.
>
> [2] Yazdani-Jahromi, Mehdi, et al. "HELM: Hierarchical Encoding for mRNA Language Modeling." ICLR, 2025.
>
> [3] Outeiral, Carlos et al. "Codon language embeddings provide strong signals for protein engineering." Nature Machine Intelligence, 2024.
>
> [4] Yazdani-Jahromi, Mehdi, et al. "Protein Translation Equivariant Encoding for mRNA Language Models." NeurIPS, 2025.
>
> [5] Wood, Matthew et al. "Helix-mRNA: A Hybrid Foundation Model For Full Sequence mRNA Therapeutics." arXiv, 2025.

---

> ### Author Response · Authors · 2025-11-20
> **Response to Review-Response 3/3**
>
> We believe the revisions and additional experiments directly address all points raised and significantly improve the clarity and rigor of the work. If the reviewer concurs, we kindly request consideration of an updated score!

---

> > ### Author Response · Authors · 2025-11-26
> > **Friendly request for feedback...**
> >
> > Dear reviewer,
> >
> > We kindly ask for a feedback regarding our new experiments, detailed explanations and revised version of the paper. We would like to understand if your concerns are now adequately addressed and if you would kindly consider raising your score in light of this. Additionally, if you would like to voice additional concerns or add details to any of the discussed points, we would be delighted to incorporate them in the paper in adequate form as well. We want to thank you in advance for your time and efforts!

---

> > > ### Comment · Reviewer_5EoM · 2025-11-27
> > >
> > > Dear authors,
> > >
> > > Thank you for thoughtfully addressing most of my earlier concerns and for including additional baselines that improve the presentation of the results. I also appreciate your commitment to ensuring reproducibility upon acceptance.
> > >
> > > I still have several unaddressed concerns that I would appreciate further clarification on:
> > >
> > > **Downstream task results.**
> > > 1. The results in Table 1 are listed as single values, missing standard deviation and confidence intervals. It is also unclear whether these correspond to mean results or single runs. This aligns with the first comment from Reviewer Vc4D and would be helpful to address for transparency.
> > > 2. In the rebuttal, you mentioned using "public pre-defined dataset splits," and that "Ab1 and Ab2 splits come from HELM." However, by inspecting the HELM paper [1], it can be noticed that different values are reported for the Ab1 and Ab2 downstream tasks than were listed in the proposed paper ($ 0.767 $ vs $ 0.714 $ and $ 0.609 $ vs $ 0.548 $). Also, for the COVID-19 dataset, the reported result is $ 0.775 $, whereas in [1] it was $ 0.833 $. The HELM paper also states that they used predefined splits for fair comparison. Could you please clarify the source of these differences?
> > > 3. Several baseline results are missing from Table 1 and Table 4 in Appendix. This was explained by "Because of the large sizes of the foundation models, not all experiments were feasible due to hardware constraints." I still believe it is important to have these results.
> > > 4. In the appendix, it was mentioned that "baselines were pretrained on different corpora, making the comparison less relevant than the comparison shown in Table 1." However, models like mRNA-FM and AIDO.RNA-1.6B-CDS in Table 4 were pretrained/finetuned on CDS data, while RNA-FM in Table 1 was pretrained only on ncRNA. Given this, I think it would actually be useful and more relevant to move some of the results from Table 4 to the main paper for a fairer comparison.
> > > ---
> > > [1] Yazdani-Jahromi, Mehdi, et al. "HELM: Hierarchical Encoding for mRNA Language Modeling." ICLR, 2025.

---

> ### Author Response · Authors · 2025-11-28
> **Addressing all questions from reviewer on downstream task results point-by-point 1/2**
>
> We thank the reviewer for their response. We address the remaining concerns below.
>
> **1) Performance variance:** Due to the fact that we use predefined splits from prior works ([1], [2], [3], [4]),  we cannot directly compute a variance, since there are no additional splits for such experiments. We could define our own new splits, but this would reduce comparability to existing work and require significantly more compute. On top of that, the results from future work would not be comparable unless they adhere to our newly introduced splits, potentially hindering future research.
>
> While variance estimates cannot be computed under fixed splits following prior works, the magnitude and consistency of improvements across datasets provide strong evidence that our gains are not due to chance. Across the datasets where baselines exist, our best hyperbolic models achieve substantial relative improvements over the strongest competing non-hierarchical models: \
> ***+13.3% on Ab1, \
> +51.5% on Ab2, \
> +7.7% on mRFP, \
> +2.2% on COVID-19, \
> +4.4% on Fungal, \
> +4.8% on iCodon, \
> and on E. coli, our model matches the best baseline and still improves +7.1% over EVO-2.***
>
> These improvements are not small fluctuations but **consistent performance gains across nearly all datasets**. Under fixed-split evaluation, such systematic improvements across heterogeneous datasets serve as strong empirical evidence that our HyperHELM provides a consistent advantage over baselines. Also note that our model performs better than the additional baselines despite having far less parameters than, for example mRNA-FM (5x), AIDO.RNA-1.6B-CDS (32x) and EVO 2 (140x).
>
> **2) Discrepancy in HELM performance:** There is a bug in the official public HELM implementation, raised in this GitHub issue from two months ago: [https://github.com/johnsonandjohnson/HELM/issues/1](https://github.com/johnsonandjohnson/HELM/issues/1). To obtain the numbers that we report for HELM, we use the official implementation, but where we fix this bug. The resulting numbers come out lower than those reported in the HELM paper for some datasets.
>
> **3) Missing values for some baselines:** Given our hardware restrictions, it is not possible for us to obtain all missing values. Note that the additional requested baselines are significantly larger than the models included in the original paper, and they operate with much larger representations. As such, training and even just evaluating these billion-scale foundation models requires huge amounts of compute and memory, which we do not have. At the same time, the results that we do have already indicate that our model performs competitively despite having far fewer parameters (50M) than the much larger foundation models: mRNA-FM (5x parameter count), AIDO.RNA-1.6B-CDS (32x) and EVO 2 (140x).
>
> More importantly, the **primary goal of this work is to explore hierarchical learning of bio-language models through the lens of hyperbolic geometry**. Therefore, HyperHELM is not proposed as a new, general-purpose mRNA foundation model intended to supersede all existing large-scale models, but it is proposed as a principled hyperbolic extension of a hierarchical bio-language modeling framework. For this purpose, the most informative and direct comparison is against the hierarchical Euclidean baseline (HELM), since this is the only regime where model size, architecture, and pre-training data are matched, and the effect of our methodological contribution is isolated. In this controlled setting, HyperHELM consistently outperforms its Euclidean counterpart, and the ablation with hyperbolic MLR further shows that the improvements do not come from a naïve “switch to hyperbolic,” but specifically from the non-trivial hyperbolic prototype formulation to MLM pre-training.
>
> In light of this, we ask the reviewer to also assess our work through the lens of its methodological advance in hyperbolic bio-language modeling and its controlled comparison against the Euclidean baseline, rather than as a direct parameter-for-parameter competitor to all existing mRNA foundation models, including foundational models that are over 100× larger, such as EVO-2. The central scientific question addressed is whether the hyperbolic approach improves over the Euclidean one in the context of mRNA language modeling, and the presented results clearly support this.
>
> **3) Include results for baselines in main text Table 1 instead of in Appendix:** We are happy to comply with the reviewer's request and have moved the mRNA baselines (which have been trained on mRNA corpora similar to our HyperHELM) from the Appendix tables to the main text Table 1 in the paper correspondingly and highlighted this change in red in Table 1.
>
> Since we have addressed all your questions in the original review as well as in follow-up with extensive experiments and revision, we kindly request the reviewer to consider raining their score!

---

> ### Author Response · Authors · 2025-11-28
> **Addressing all questions from reviewer on downstream task results point-by-point 2/2**
>
> Additional references:
>
> [1] Yazdani-Jahromi, Mehdi, et al. "HELM: Hierarchical Encoding for mRNA Language Modeling." ICLR, 2025. \
> [2] Outeiral, Carlos et al. "Codon language embeddings provide strong signals for protein engineering." Nature Machine Intelligence, 2024. \
> [3] Wood, Matthew et al. "Helix-mRNA: A Hybrid Foundation Model For Full Sequence mRNA Therapeutics." arXiv, 2025. \
> [4] Li, Sizhen, et al. "CodonBERT: Large Language Models for mRNA design and optimization." Genome Research, 2023.

---

### Official Review · Reviewer_Vc4D · 2025-10-26

**Soundness:** 2
**Presentation:** 2
**Contribution:** 2
**Rating:** 2
**Confidence:** 3

**Summary:**

The authors propose HyperHELM, a framework for representing mRNA sequences in hyperbolic space to better capture hierarchical structure that may be present in the data. HyperHELM uses a Euclidean transformer backbone paired with several heads for MLM prediction: hyperbolic multinomial logistic regression, and two prototype-based classifiers that use either geodesic or entailment cone energy as similarity functions. The codon-amino acid hierarchy is used to generate fixed prototypes for the latter options. HyperHELM compares against several RNA foundation models, and against a fully Euclidean model trained on the same data. The authors find performance benefits on 9/10 mRNA property prediction tasks when both the prototype and hyperbolic representation learning methods are used in conjunction. The authors conclude by analyzing the performance of HyperHELM as a function of sequence length and GC content on several downstream tasks.

**Strengths:**

- To my knowledge, this work is the first implementation of hyperbolic learning into mRNA language modelling. The adaptation of new learning approaches to mRNA biology is an area of high significance.
- The use of prototype learning to encode the codon-amino acid hierarchy has strong biological relevance and offers an interpretable inductive bias for mRNA representation learning in the CDS region.
- The evaluation across sequence length and GC content distributions reveals biologically meaningful insights about model generalization.
- Presentation of background and methodology on hyperbolic learning is written with high clarity.

**Weaknesses:**

- My primary concern lies with biological validity of the tokenization strategy underlying the prototype learning. While using trinucleotide tokenization in the CDS region makes sense, the paper does not explain how the reading frame is selected. Without proper alignment to the annotated start codon, the tokenization could be out-of-frame, preventing the codon-amino acid structure from being leveraged. Furthermore, this strategy has no biological validity in an mRNA's UTR regions. For example, the presence of upstream AUGs [1] strongly affect mRNA translational efficiency, but the proposed tokenization strategy could completely fail to represent them due to misalignment.

- Pre-training dataset description is unavailable without referencing an external paper, hurting self-contained clarity. More critically, if I am not mistaken, the pre-training database primarily contains antibody-encoding sequences. Given that this corpus is relatively specialized, it introduces a confound for the antibody-related downstream tasks, as domain-matching the pre-training data may drive a significant portion of the model improvement over language model baselines.

Experimental evaluation is relatively weak.
- Choice of baseline foundation models is incomplete. Several models identified in the related works are not evaluated, in particular models which are actually suited for mRNA biology. Current selection include models trained on ncRNA (RNA-FM) and pre-mRNA (SpliceBERT) while models such as mRNA-FM, AIDO.RNA-CDS, Orthrus, Evo2, Helix-mRNA, etc. all provide stronger and more appropriate baselines.
- Selected tasks almost exclusively focus on quantifying protein expression, while mRNA biology encompasses many additional properties. If HyperHELM is meant to only focus on expression, the scope of the claims should be restricted.

Incomplete related work coverage:
- Critically, the paper should make specific comparison to "Hyperbolic Genome Embeddings" [2], which previously explored hyperbolic learning in genomic sequences. It may be appropriate to restrict the scope of the originality claims and suggested future directions in response to this citation.
- The related works section contains an incomplete overview of methods for mRNA representation learning. Several of the strongest models such as AIDO.RNA-CDS [3], Orthrus [4], and LoRNASH [5] are not referenced. Supervised approaches that are directly applicable to the downstream tasks such as RiboNN [6] and Optimus 5' [1] are not mentioned.

[1]: https://www.nature.com/articles/s41587-019-0164-5
[2]: https://arxiv.org/abs/2507.21648
[3]: https://www.biorxiv.org/content/10.1101/2024.11.28.625345v1
[4]: https://www.biorxiv.org/content/10.1101/2024.10.10.617658v3
[5]: https://www.biorxiv.org/content/10.1101/2024.08.26.609813v2
[6]: https://pubmed.ncbi.nlm.nih.gov/39149337/

**Questions:**

I would appreciate a rebuttal to the above concerns, particularly a detailed description of how mRNA transcripts were processed for pre-training, and how the reading frame problem was tackled. Specifically, my questions are:
- Are pre-training sequences full transcript, or CDS-only?
- How was the reading frame determined?
- How are UTRs handled?

Furthermore, I am curious whether the proposed downstream tasks empirically contain hyperbolicity. I would appreciate the authors considering whether an evaluation of the downstream tasks as proposed in Hyperbolic Genome Embeddings would be suitable here.

---

> ### Author Response · Authors · 2025-11-20
> **Response to review- Response 1/4**
>
> We thank the reviewer for their positive remarks regarding the novelty, interpretability and significance of our approach and for their compliments regarding the writing. Below we have addressed each of the reviewer's concerns and we have updated the manuscript accordingly.
>
> **Alignment to mRNA reading frame.**
> We would like to clarify that we use the pre-training data directly from HELM [1] which already comes with annotations for CDS region. The codon–amino acid hierarchy that underlies our prototype construction is indeed most biologically meaningful within CDS regions. In practice, many mRNA design and analysis tasks, such as codon optimization [2], expression tuning [3], and stability engineering for therapeutic constructs [4], focus predominantly on the CDS, where synonymous codon choices directly modulate mRNA properties.
>
> Crucially, because of the central importance of the CDS, there now exists a mature ecosystem of tools for reliably identifying reelvant open reading frames (ORFs) and CDS segments in mRNAs. These tools include annotation-based CDS definitions in curated databases [5], ORF prediction tools that scan for start/stop codons under organism-specific constraints [6,7], and gene-structure predictors that leverage coding potential and splice-aware models [8]. As a result, in typical downstream applications, CDS boundaries and reading frames are known a priori, and sequences can be provided to our model already aligned to the correct frame. In this realistic deployment setting, where the input is a verified CDS rather than an unannotated full transcript, our codon-level tokenization is naturally in-frame, and systematic misalignment is therefore not expected to be an issue. We added clarification about identifying the CDS region in the main text of the paper (lines 322 and 366-369).
>
> At the same time, we agree with the reviewer that the codon-centric formulation may not capture the full complexity of transcript, including UTR regions, which pose their own modeling challenges addressed by specialized UTR models [9]. In this sense, our HyperHELM is complementary to such specialized methods and could be integrated with dedicated UTR models in future work. In this work, we focus purely on mRNA regions without UTR as is the case with many other recent works in the field [1,11,12,13].
>
> **Pre-training dataset.**
> We appreciate the reviewer’s focus on the self-contained clarity of the paper. In the revised version, we have explicitly added the pre-training corpus details in Appendix C.
>
> Regarding the concern about a potential confound from the pre-training data, we agree that the composition of the pre-training corpus influences absolute performance. For this reason, our central methodological comparison is against Transformer XE and HELM baselines, which are _pre-trained on exactly the same dataset_ as our model. These baselines, therefore, provide a direct comparison that isolates the effect of our proposed hyperbolic approach from any properties of the pre-training corpus itself. In other words, any performance difference between Transformer XE, HELM and our model cannot be attributed to domain-matching, as all three models see the same sequences during pre-training.
>
> At the same time, we chose to also include external foundation models (RNA-FM, SpliceBERT, CodonBERT from earlier and the newly added Helix-mRNA, mRNA-FM, AIDO.RNA-CDS, and EVO 2) into our experimental comparison to give a broader view of how HyperHELM compares to existing RNA/mRNA representation learning methods, even though we cannot control or fully harmonize their pre-training data.
>
> **Task diversity.**
> Our experiments include degradation prediction (COVID-19), thermostability prediction (iCodon), transcript abundance prediction (P. pastoris), and protein abundance prediction (D. melanogaster and S. cerevisiae). The latter differs from protein expression as protein abundance reflects the combined effects of transcription, translation efficiency, and degradation, and is typically derived from proteomics measurements rather than a single reporter assay [10]. With this, we respectfully disagree with the reviewer that our experiments are exclusively limited to protein expression, as 5 out of 10 datasets involve predicting properties that are different from standalone protein expression.

---

> ### Author Response · Authors · 2025-11-20
> **Response to Review-Response 2/4**
>
> **Comparison against additional baselines.**
> We thank the reviewer for suggesting additional baselines. Based on their suggestions, we have included several additional baselines, namely EVO 2 (7B), Helix-mRNA, mRNA-FM, and AIDO.RNA-CDS. The results are shown in the table below. Due to the computational and time constraints, we are not able to obtain results of all models on all downstream datasets. As can be seen, our model outperforms the additional baselines, including EVO 2, on all of the datasets except for AIDO.RNA-CDS on COVID-19, where our model performs on par, and mRNA-FM on _E. coli_ where mRNA-FM is the strongest model. These additional results have been included in the paper. Missing value for Fungal dataset with AIDO.RNA-CDS indicates the runs did not execute due to hardware limitations. We have also included these in Appendix Section G.
>
>
> | Dataset       | Helix-mRNA |  mRNA-FM | EVO 2 | AIDO.RNA-CDS | Proto Dist. (Ours)     | Proto Entail. (Ours)|
> |---------------|:----------:|:--------:|:-----:|:------------:|:----------------------:|:------------------------:|
> | Ab1           |    0.535   |   0.656  | 0.129 |     0.663    |    0.713               |   **0.751**              |
> | Ab2           |    0.283   |   0.373  | 0.141 |     0.398    |  **0.575**             |     0.569                |
> | mRFP          |    0.432   |   0.739  | 0.239 |     0.787    |  **0.819**             |     0.802                |
> | COVID-19          |    0.643   |   0.762  | 0.386 |     0.804    |    0.785               |   **0.807**              |
> | Fungal        |    0.689   |   0.722  | 0.400 |              |    0.735               |   **0.741**              |
> | Ecoli         |    40.0    | **53.3** |  40.0 |     52.9         |     50.8               |      48.4                |
>
> **Comparison to hyperbolic genome embeddings.**
> We appreciate the reviewer’s suggestion to compare to Hyperbolic Genome Embeddings (HGE). While both HGE and HyperHELM explore the use of hyperbolic geometry for biological data, the settings and goals differ in several important ways. Firstly, HGE is a DNA model while HyperHELM specifically focuses on mRNA. Secondly, HGE proposes a fully hyperbolic architecture trained directly on specific downstream tasks, whereas HyperHELM is designed as a self-supervised language model for large-scale pre-training on unlabeled mRNA sequences, with the goal of learning general-purpose representations that can be reused across many downstream tasks. Consequently, our primary conceptual and experimental comparisons are to other bio-language models that perform MLM-style pre-training, rather than to task-specific supervised architectures. This focus on large-scale pre-training also drives key technical choices that differ from HGE: in particular, we find that hyperbolic prototype-based heads are crucial for MLM performance, while hyperbolic MLR alone underperforms, and we adopt a projection-based approach (Euclidean backbone with hyperbolic heads) rather than a fully hyperbolic network to maintain scalability and compatibility with standard transformer pre-training.
>
> We also would like to point out that our originality claims are intended to be restricted to the use of hyperbolic geometry for pre-training an mRNA language model, not to hyperbolic learning on biological sequences in general.
>
> At the same time, we agree with the reviewer that HGE is an important early work in hyperbolic learning for genomic sequences, and we have now explicitly cited it and discussed it in the revised paper, clarifying both the connections and the differences in scope and objectives in lines 152-154 and Appendix Section J.
>
>
> **Additions to related work.**
> We thank the reviewer for pointing out additional relevant work on mRNA representation learning and supervised models directly applicable to our downstream tasks. We agree that methods such as AIDO.RNA-CDS, Orthrus, LoRNASH, RiboNN, and Optimus 5' are important contributions to this area. In the revised version, we have expanded the related work section to explicitly discuss these models (lines 120-139).
>
>
> **Empirical hyperbolicity of datasets.**
> We would like to point out that we generate embeddings using a model that is trained to output representations in hyperbolic space. As such, the hyperbolicity measure will be heavily biased towards low $\delta$-hyperbolicity. Therefore, we believe that computing such values may result in a misleading overestimation of the potential hyperbolicity of such datasets. Moreover, $\delta$-hyperbolicity is a tool developed for determining the hyperbolicity of a metric space [14]. It is unclear how reliably one can estimate it from a finite sample drawn from a distribution over that space. For these reasons, we are cautious about drawing strong conclusions from such measurements.

---

> ### Author Response · Authors · 2025-11-20
> **Response to Review-Response 3/4**
>
> **Reviewer's questions:**
> - Our model is pre-trained on CDS-only sequences. The pre-training corpus obtained directly from prior work HELM [1] already contains annotations for the relevant open reading frame.
> - In all downstream experiments, the model is likewise applied only to the CDS region of each construct as is done in other works [1, 11, 12 13]. CDS boundaries come either directly from existing annotations or can be obtained via standard ORF-identification tools. UTR segments are not provided as input to HyperHELM in our experiments similar to prior works [1, 11, 12 13].
>
> We appreciate the reviewer’s focus on clarity; we have made these points explicit in the main text and appendix in the revised version.
>
> **References:**
>
> [1] Yazdani-Jahromi et al. HELM: Hierarchical Encoding for mRNA Language Modeling, ICLR 2025.
>
> [2] Zhang et al. Algorithm for optimized mRNA design improves stability and immunogenicity. Nature 2023.
>
> [3] Barrington et al. Synonymous codon usage regulates translation initiation. Cell reports 2023.
>
> [4] Presnyak et al. Codon optimality is a major determinant of mRNA stability. Cell 2015.
>
> [5] O'Leary et al. Reference sequence (RefSeq) database at NCBI: current status, taxonomic expansion, and functional annotation. Nucleic acids research 2016.
>
> [6] Hyatt et al. Prodigal: prokaryotic gene recognition and translation initiation site identification. BMC bioinformatics 2010.
>
> [7] Urminder Singh, Eve Syrkin Wurtele. Orfipy: a fast and flexible tool for extracting ORFs. Bioinformatics 2021.
>
> [8] Gabriel et al. BRAKER3: Fully automated genome annotation using RNA-seq and protein evidence with GeneMark-ETP, AUGUSTUS, and TSEBRA. Genome research 2024.
>
> [9] Chu et al. A 5′ UTR language model for decoding untranslated regions of mRNA and function predictions. Nature Machine Intelligence 2024.
>
> [10] Christine Vogel and Edward M. Marcotte. Insights into the regulation of protein abundance from proteomic and transcriptomic analyses. Nature reviews genetics 2012.
>
> [11] Li, Sizhen, et al. "CodonBERT: Large Language Models for mRNA design and optimization." Genome Research, 2023.
>
> [12] Yazdani-Jahromi, Mehdi, et al. "Protein Translation Equivariant Encoding for mRNA Language Models." NeurIPS, 2025.
>
> [13] Wood, Matthew et al. "Helix-mRNA: A Hybrid Foundation Model For Full Sequence mRNA Therapeutics." arXiv, 2025.
>
> [14] Gromov, Mikhael. "Hyperbolic groups." Essays in group theory. New York, NY: Springer New York, 1987. 75-263.

---

> ### Author Response · Authors · 2025-11-20
> **Response to Review-Response 4/4**
>
> Given our detailed clarifications, experimental comparisons with more baselines and related works, we hope to have addressed the reviewer's concerns. If the reviewer agrees, we would appreciate them updating their current score!

---

> > ### Author Response · Authors · 2025-11-26
> > **Friendly request for feedback...**
> >
> > Dear reviewer,
> >
> > We kindly ask for a feedback regarding our new experiments, detailed explanations and revised version of the paper. We would like to understand if your concerns are now adequately addressed and if you would kindly consider raising your score in light of this. Additionally, if you would like to voice additional concerns or add details to any of the discussed points, we would be delighted to incorporate them in the paper in adequate form as well. We want to thank you in advance for your time and efforts!

---

> > > ### Comment · Reviewer_Vc4D · 2025-11-27
> > >
> > > Thank you for the extensive additional experiments; the majority of my concerns have been addressed and I have raised my score accordingly.
> > >
> > > I have a few remaining concerns:
> > >
> > > - Since the performance gaps between several models are smaller in the revised manuscript, it would be helpful if an estimation of performance variance (e.g. standard deviation, confidence interval) was included.
> > > - It would be helpful to report the performance of supervised baselines (e.g. Euclidean CNN, Hyperbolic CNN) on the evaluation tasks. Although the current results demonstrate the benefits of pre-training a CDS FM using a hyperbolic objective, this proposed comparison would better support the utility of HyperHELM as a general method to model translation / stability / etc.
> > > - The choice to model only CDS regions is well-motivated by real-world use cases such as CDS optimization to boost translation. However, the manuscript is terminologically misleading in places: it uses “mRNA” where “CDS” is meant, and is written as if the findings apply to full mRNA transcripts, even though HyperHELM excludes UTRs. For example, the abstract states that “hyperbolic geometry [is] an effective inductive bias for hierarchical language modelling of mRNA sequences”; if “mRNA sequences” is interpreted as full transcripts, this claim is not supported by the experiments presented. It is entirely possible that applying the hyperbolic objective to full mRNAs could be detrimental, since codon structure is not biologically relevant in UTRs. Revising the manuscript to make the CDS-only scope explicit from the outset would improve clarity and avoid over-generalization.

---

> ### Author Response · Authors · 2025-11-28
> **Addressing all mentioned concerns with revision and additional experiments of supervised baselines (Euclidean CNN, Hyperbolic CNN)**
>
> We thank the reviewer for raising their score and address their remaining concerns below.
>
> **1) Performance variance:** Due to the fact that we use predefined splits from prior works ([1], [2], [3], [4]),  we cannot directly compute a variance, since there are no additional splits for such experiments. We could define our own new splits, but this would reduce comparability to existing work and require significantly more compute. On top of that, the results from future work would not be comparable unless they adhere to our newly introduced splits, potentially hindering future research.
>
> While variance estimates cannot be computed under fixed splits following prior works, the magnitude and consistency of improvements across datasets provide strong evidence that our gains are not due to chance. Across the datasets where baselines exist, our best hyperbolic models achieve substantial relative improvements over the strongest competing non-hierarchical models: \
> ***+13.3% on Ab1, \
> +51.5% on Ab2, \
> +7.7% on mRFP, \
> +2.2% on COVID-19, \
> +4.4% on Fungal, \
> +4.8% on iCodon, \
> and on E. coli, our model matches the best baseline and still improves +7.1% over EVO-2.***
>
> These improvements are not small fluctuations but **consistent performance gains across nearly all datasets**. Under fixed-split evaluation, such systematic improvements across heterogeneous datasets serve as strong empirical evidence that our HyperHELM provides a consistent advantage over baselines. Also note that our model performs better than the additional baselines despite having far less parameters than, for example mRNA-FM (5x), AIDO.RNA-1.6B-CDS (32x) and EVO 2 (140x).
>
> **2) Performance of supervised baselines:**  To address the reviewer’s suggestion, we trained supervised baselines using the exact same prediction head architecture used on HyperHELM: a Euclidean TextCNN and a Hyperbolic TextCNN. This isolates the effect of the representation space (Euclidean vs. hyperbolic) separately from the benefits of large-scale CDS pretraining.
> The results below show that both supervised CNN baselines perform substantially worse than our hyperbolic ProtoDist and ProtoEntail models across all datasets, often by large margins. In particular: \
> ***HyperHELM’s ProtoDist and ProtoEntail outperform the best CNN baseline
> by 41–74% on antibody datasets (Ab1, Ab2), \
> by >70% on mRFP, \
> by ~35% on COVID-19, \
> by ~+27% on E.Coli, \
> and by >300% on long-sequence eukaryotic datasets (D. melanogaster, S. cerevisiae, P. pastoris) over both Euclidean and hyperbolic CNNs.***
>
> These results confirm that the gains we observe are not simply due to using a prediction head, but instead arise from the our HyperHELM large-scale CDS pre-training. Supervised hyperbolic CNNs alone do not achieve strong performance, whereas HyperHELM consistently does.
>
> For completeness, we provide the core comparison table below, and also include it in Appendix Table 4 in Appendix Section G (highlighted in red in revised paper).
>
> | Dataset          | Euclidean CNN | Hyperbolic CNN | HyperHELM Proto Dist. (Ours) | HyperHELM Proto Entail. (Ours)|
> |------------------|----------------|-----------------|--------------|----------------|
> | Ab1              | 0.421          | 0.518           | 0.713        | **0.751**      |
> | Ab2              | 0.243          | 0.252           | **0.575**    | 0.569          |
> | mRFP             | 0.474          | 0.193           | **0.819**    | 0.802          |
> | COVID-19         | 0.602          | 0.480           | 0.785        | **0.807**      |
> | D. melanogaster  | 0.103          | 0.037           | 0.394        | **0.450**      |
> | S. cerevisiae    | 0.143          | 0.121           | **0.434**    | 0.397          |
> | P. pastoris      | 0.301          | 0.229           | **0.676**    | 0.671          |
> | Fungal           | 0.606          | 0.580           | 0.735        | **0.741**      |
> | *E. coli*        | 40.0           | 40.0            | **50.8**     | 48.4           |
> | iCodon           | 0.152          | 0.143           | 0.535        | **0.539**      |
>
>
> **3) Clarity regarding the use of full mRNA transcripts versus CDS regions.** We agree with the reviewer and have changed the writing accordingly and highlighted the changes in red in abstract and introduction of revised paper.
>
> Given that we have addressed all your concerns with extensive experiments and revision, we kindly request the reviewer to consider raining their score!
>
> [1] Yazdani-Jahromi, Mehdi, et al. "HELM: Hierarchical Encoding for mRNA Language Modeling." ICLR, 2025. \
> [2] Outeiral, Carlos et al. "Codon language embeddings provide strong signals for protein engineering." Nature Machine Intelligence, 2024. \
> [3] Wood, Matthew et al. "Helix-mRNA: A Hybrid Foundation Model For Full Sequence mRNA Therapeutics." arXiv, 2025. \
> [4] Li, Sizhen, et al. "CodonBERT: Large Language Models for mRNA design and optimization." Genome Research, 2023.

---

### Official Review · Reviewer_Ak6U · 2025-10-29

**Soundness:** 3
**Presentation:** 2
**Contribution:** 3
**Rating:** 6
**Confidence:** 2

**Summary:**

This paper introduces HyperHELM, a framework for modeling mRNA sequences. The authors argue that the inherent hierarchical structure of mRNA is not well represented by the Euclidean geometry used in standard language models. They propose a hybrid model that combines an efficient Euclidean transformer backbone with a hyperbolic prediction head. This model is pre-trained using a masked language modeling objective directly in hyperbolic space. The hyperbolic representations are guided by prototypes derived from the known codon hierarchy. The paper evaluates two types of prototype classifiers, one based on distance and another on entailment cones. The results show that HyperHELM outperforms Euclidean baselines. The model also shows better out-of-distribution generalization to long sequences and sequences with low GC content.

**Strengths:**

1. Novel and well motivated application: The paper applies hyperbolic geometry to mRNA language modeling. This is a very natural and well justified choice.

2. The choice of a hybrid architecture is pragmatic. The model retains the hardware efficiency of a standard Euclidean transformer for most of its computations. It only projects into hyperbolic space at the final layer. This design gains the benefits of hyperbolic geometry without incurring the full computational cost of a purely hyperbolic network.

3. The experiments are well designed. The authors compare HyperHELM not just to standard baselines but also to HELM, a state of the art Euclidean model that is also hierarchy aware. This comparison effectively isolates the benefit of using hyperbolic geometry rather than just using the hierarchy itself.

**Weaknesses:**

1. The prototypes used in the hyperbolic head are fixed. They are generated once from the codon hierarchy using an embedding method and are not updated during training. It seems like a potentially missed opportunity. Allowing the prototypes to be learnable might permit the model to fine tune the hierarchical relationships based on the data.

2.  The paper focuses exclusively on masked language modeling (MLM). While this is a standard objective, it would be interesting to see if the benefits of the hyperbolic hierarchy encoding also apply to generative tasks, such as those using a causal language modeling objective.

3.  The HyperHELM MLR variant performed very poorly, even worse than the Euclidean baselines. The paper could benefit from a more detailed discussion of why this specific formulation failed, while the prototype based methods were so successful. It strongly suggests that the prototype learning aspect is just as critical as the hyperbolic space itself.

**Questions:**

1. Could the authors elaborate on the poor performance of the HyperHELM MLR variant? What is the intuition for why this method struggles so much, given that the prototype models demonstrate a clear benefit from the hyperbolic space?

2. The model architecture involves a projection from the Euclidean backbone's output to the hyperbolic space. Did the authors study the sensitivity of the model to this projection step? For instance, how much does the choice of projection function (detailed in Equation 9) matter for the final performance?

3. The paper uses the HS DTE method to embed the codon hierarchy and generate the fixed prototypes. How critical is this specific embedding choice to the model's success? Have the authors considered or briefly tested other, perhaps simpler, tree embedding methods?

---

> ### Author Response · Authors · 2025-11-20
> **Response to review- Response 1/2**
>
> We thank the reviewer for their compliments regarding our novel and well-motivated method and the design of the experiments. Moreover, we are grateful for the constructive feedback and new suggestions!
>
> **Learnable prototypes.**
> We initially opted for fixed prototypes to ensure the hyperbolic prototype learning scheme's effect could be observed in isolation, without the influence of too many other factors. However, similarly to the reviewer, we also wondered whether learnable prototypes could lead to further improvements. To test this, we made the prototypes learnable, initialized using HS-DTE, and performed pre-training following the same setting as in the paper. For the prototypes, we use the same learning rate and scheduler as before, but with Riemannian SGD [1]. The results are shown in the table below. As can be seen, learnable prototypes led to further improvements on several downstream tasks. Both distance and entailment based learnable protoype models improved upon their fixed protype counterparts on 6 out of 9 datasets (Spearman rank corelation for regression and accuracy for classification tasks are reported). These results indicate the potential of the learnable prototypes approach, and further improve our results and paper. We thank the reviewer for their suggestion, and we have added these results and the discussion into the paper (Table 1 main text and Appendix C).
>
>
> | Dataset       | Fixed Proto Dist. | Fixed Proto Entail. | Proto Dist. learnable | Proto Entail. learnable |
> |---------------|:-----------------:|:-------------------:|:---------------------:|:-----------------------:|
> | Ab1           |    0.713          |   **0.751**         |         0.743         |          0.736          |
> | Ab2           |    0.575          |     0.569           |       **0.603**       |          0.589          |
> | mRFP          |    0.819          |     0.802           |         0.800         |        **0.820**        |
> | COVID-19      |    0.785          |     0.807           |       **0.822**       |        **0.822**        |
> | Fungal        |    0.735          |     0.741           |         0.742         |        **0.754**        |
> | E. Coli       |     50.8          |      48.4           |        **53.0**       |           50.9          |
> | iCodon        |    0.535          |     0.539           |       **0.545**       |          0.536          |
> | ppastoris     |  **0.676**        |     0.671           |         0.672         |          0.672          |
> | dmelanogaster |    0.394          |   **0.450**         |         0.442         |          0.447          |
> | scerevisiae   |    0.434          |     0.397           |         0.424         |        **0.480**        |
>
>
> **Focus on MLM.**
> We chose to focus on the MLM setting, which is the standard paradigm in many recent bio-sequence modeling work (e.g., ESM-2, ESM-3, RNA-FM, CodonBERT). While CLM and, more broadly, generative modeling are indeed compelling future directions, extending them to the hyperbolic setting introduces several additional non-trivial challenges. In particular, autoregressive objectives require sequential predictions in hyperbolic space, careful handling of curvature-aware normalization, and stability considerations when repeatedly applying exponential and logarithmic maps during generation. Addressing these complexities is an exciting direction for future work, but is beyond the scope of our current study. We view exploring hyperbolic CLM and generative models as a non-trivial extension and a valuable follow-up paper in its own right.
>
> **The poor performance of MLR.**
> As the reviewer notes, the MLR variant performs poorly, which underlines that simply switching to hyperbolic geometry is not enough. Our hypothesis regarding this observation is most evident when tracking hyperbolic norms of representations during training (added Figure 5 in Appendix Section F). Essentially, standard MLR tends to drive the embeddings away from the origin, resulting in quickly growing hyperbolic norms of the representations [9]. In hyperbolic space, large norms quickly lead to numerical instability, and hyperbolic learning libraries such as HypLL handle this by clipping points back into a bounded region. As training progresses, MLR therefore produces heavily clipped representations, effectively discarding information encoded in the norm and hindering learning. The hyperbolic prototypes approach, however, is not prone to ever-growing representation norms and hence requires no clipping, which leads to increased representation flexibility and stability. We have added the detailed explanation in Appendix F.

---

> ### Author Response · Authors · 2025-11-20
> **Response to review- Response 2/2**
>
> **The choice of projection function.**
> In hyperbolic literature, the typical method for projection to hyperbolic space is the exponential map, with the base point being set to the origin of the hyperbolic space. We are not aware of any other commonly used projections, but if the reviewer has any in mind, then we would be happy to discuss their relevance to the method. This choice of base point for the exponential map is well-established in the hyperbolic learning literature [2–8], as it greatly simplifies the exponential map and has been shown empirically to have barely any impact on performance [4]. Moreover, in our setting the origin has an important geometric meaning. Namely, it represents the origin of the tree. Therefore, setting the base point to the origin essentially centers token representations around the root of the tree, which intuitively seems to be a sensible approach.
>
> We have now also experimented with the idea of using a learnable base point with fixed entailment prototypes. Our origin choice performs better on 4/9 datasets and on par on another 4/9, validating our initial design in the paper (now added to Appendix L).
>
> | Dataset | Learnable | Origin |
> |---------|-----------|--------|
> | Ab1 | 0.701 |  **0.751** |
> | scerevisiae | 0.369 |  **0.397** |
> | COVID-19 | 0.783 |  **0.807** |
> | Fungal | 0.724 |  **0.741** |
> | EColi |  **50.6** | 48.4 |
> | dmelanogaster |  **0.451** |  **0.450** |
> | Ab2 | **0.570** |  **0.569** |
> | mRFP | **0.805** |  **0.802** |
> | ppastoris |  **0.671** |  **0.671** |
>
>
>
> **The importance of the embedding method.**
> We used the HS-DTE method as it is obtains the lowest distortion and because it is fast compared to the available optimization-based methods [9]. The first commonly used method for generating tree embeddings is the Poincaré embeddings method [10], which is generally a bit slower and results in less faithful embeddings. However, when using this method to generate our prototypes, we found that our method is not particularly sensitive to this choice as can be seen in the table below. Hence, we conclude that our model can also be integrated with simpler tree embedding methods such as Poincaré embeddings. This analysis has also been added in Appendix H now.
>
> | Dataset | Poincaré embeddings | HS-DTE |
> |---------|:-------------------:|:------:|
> | Ab1     |        0.752        |  0.751 |
> | Ab2     |        0.546        |  0.569 |
> | SARS    |        0.820        |  0.807 |
> | Fungal  |        0.728        |  0.741 |
> | mRFP    |        0.829        |  0.802 |
>
>
> We hope that our clarifications and additional experiments address the reviewer’s concerns. If so, we kindly invite the reviewer to consider updating their score!
>
> **References:**
>
> [1] Bonnabel, Silvere. "Stochastic gradient descent on Riemannian manifolds." IEEE Transactions on Automatic Control 58.9 (2013): 2217-2229.
>
> [2] Mettes, Pascal, et al. "Hyperbolic deep learning in computer vision: A survey." International Journal of Computer Vision 132.9 (2024): 3484-3508.
>
> [3] Khrulkov, Valentin, et al. "Hyperbolic image embeddings." Proceedings of the IEEE/CVF conference on computer vision and pattern recognition. 2020.
>
> [4] Ermolov, Aleksandr, et al. "Hyperbolic vision transformers: Combining improvements in metric learning." Proceedings of the IEEE/CVF conference on computer vision and pattern recognition. 2022.
>
> [5] Atigh, Mina Ghadimi, et al. "Hyperbolic image segmentation." Proceedings of the IEEE/CVF conference on computer vision and pattern recognition. 2022.
>
> [6] Van Spengler, Max, Erwin Berkhout, and Pascal Mettes. "Poincare resnet." Proceedings of the IEEE/CVF International Conference on Computer Vision. 2023.
>
> [7] Desai, Karan, et al. "Hyperbolic image-text representations." International Conference on Machine Learning. PMLR, 2023.
>
> [8] Pal, Avik, et al. "Compositional Entailment Learning for Hyperbolic Vision-Language Models." The Thirteenth International Conference on Learning Representations.
>
> [9] van Spengler, Max, and Pascal Mettes. "Low-distortion and GPU-compatible Tree Embeddings in Hyperbolic Space." Forty-second International Conference on Machine Learning.
>
> [10] Nickel, Maximillian, and Douwe Kiela. "Poincaré embeddings for learning hierarchical representations." Advances in neural information processing systems 30 (2017).

---

> > ### Author Response · Authors · 2025-11-26
> > **Friendly request for feedback...**
> >
> > Dear reviewer,
> >
> > We kindly ask for a feedback regarding our new experiments, detailed explanations and revised version of the paper. We would like to understand if your concerns are now adequately addressed and if you would kindly consider raising your score in light of this. Additionally, if you would like to voice additional concerns or add details to any of the discussed points, we would be delighted to incorporate them in the paper in adequate form as well. We want to thank you in advance for your time and efforts!

---

### Official Review · Reviewer_bM9n · 2025-11-01

**Soundness:** 3
**Presentation:** 2
**Contribution:** 2
**Rating:** 4
**Confidence:** 4

**Summary:**

This paper introduces HyperHELM, a new framework for modeling mRNA sequences that leverages hyperbolic geometry to better capture the data's inherent hierarchical structure.

**Strengths:**

-The model's core idea is to use a geometry (hyperbolic) that naturally matches the data's inherent structure (hierarchical).

-The MLR variant performed very poorly, even worse than the Euclidean models. This serves as a powerful ablation study, demonstrating that simply using hyperbolic space is not enough. The performance gain comes from the combination of hyperbolic geometry with an explicit, hierarchy-aware prototype learning scheme

**Weaknesses:**

-The model's prototypes are explicitly fixed. They are generated by an external method, HS-DTE, before training begins. This presents a significant technical bottleneck. The entire performance of the model is permanently limited by the quality of this one-time embedding.

-The model appears to be "double-dipping" on the hierarchical information, which could be redundant or even create conflicting signals

-This method involves an exponential map centered at the origin ($exp_0^c$). This is a very specific and strong assumption, treating every output token from the Transformer as a tangent vector at the origin. This step could be a significant information bottleneck, and it's not clear if this single-point-of-origin mapping is the optimal way to bridge the two geometric spaces without losing a
lot of the rich information captured by the Euclidean backbone.

**Questions:**

-Why did you choose to keep the prototypes fixed? Do you think making them learnable would allow the model to capture more subtle, data-driven relationships beyond the canonical codon hierarchy, or would it risk "overfitting" and losing the strong biological prior?

-The HyperHELM MLR variant (which used hyperbolic space but no prototypes) performed very poorly. Does this imply that hyperbolic geometry alone is insufficient and that the explicit "prototype-based" classifier is the real source of the performance gain?

-What specific geometric property of the hyperbolic space do you believe enables the model to generalize so much better to long sequences, where the Euclidean model fails?

-why there are missing values for CodonBERT on some tasks that are in the CodonBERT paper?

-why not comparing with EVO 2 - https://www.biorxiv.org/content/10.1101/2025.02.18.638918v1.abstract


-would like to increase my score if the above concerns are well addressed

---

> ### Author Response · Authors · 2025-11-20
> **Response to review- Response 1/4**
>
> We thank the reviewer for noting the importance of our prototype learning scheme and for their helpful feedback. Below, we address the weaknesses and questions raised by the reviewer.
>
> **Learnable prototypes.**
> We initially opted for fixed prototypes to ensure the hyperbolic prototype learning scheme's effect could be observed in isolation, without the influence of many other factors. However, similar to reviewer, we wondered whether learnable prototypes lead to further improvements. To verify, we made the prototypes learnable, initialized using HS-DTE, and performed pre-training following the setting in the paper. For prototypes, we use the same learning rate and scheduler as before, but with Riemannian SGD [1]. Results are shown in table below. Learnable prototypes led to further improvements on several downstream tasks. Both distance and entailment based learnable protoype models improved upon their fixed protype counterparts on 6 out of 9 datasets (Spearman correlation for regression; accuracy for classification tasks). Results indicate the potential of learnable prototypes, and further improve our results and paper. We thank the reviewer for this suggestion. We have added these results and discussion in the paper (Table 1 main and Appendix C).
>
> | Dataset       | Fixed Proto Dist. | Fixed Proto Entail. | Proto Dist. learnable | Proto Entail. learnable |
> |---------------|:-----------------:|:-------------------:|:---------------------:|:-----------------------:|
> | Ab1           |    0.713          |   **0.751**         |         0.743         |          0.736          |
> | Ab2           |    0.575          |     0.569           |       **0.603**       |          0.589          |
> | mRFP          |    0.819          |     0.802           |         0.800         |        **0.820**        |
> | COVID-19      |    0.785          |     0.807           |       **0.822**       |        **0.822**        |
> | Fungal        |    0.735          |     0.741           |         0.742         |        **0.754**        |
> | E. Coli       |     50.8          |      48.4           |        **53.0**       |           50.9          |
> | iCodon        |    0.535          |     0.539           |       **0.545**       |          0.536          |
> | ppastoris     |  **0.676**        |     0.671           |         0.672         |          0.672          |
> | dmelanogaster |    0.394          |   **0.450**         |         0.442         |          0.447          |
> | scerevisiae   |    0.434          |     0.397           |         0.424         |        **0.480**        |
>
>
> **Double-dipping.**
> We appreciate reviewer’s observation regarding potential “double-dipping” on hierarchical information. To address this, we conducted additional experiments comparing our method when paired with standard cross-entropy (XE) versus hierarchical cross-entropy (HXE). We previously observed that combining our method with standard cross-entropy led to worse performance, hence it was not included in initial submission. We have now run full ablation to verify this behavior, and the results confirm that hierarchical cross-entropy provides superior performance  in general, without introducing conflicting signals. We have added this in Appendix Section K.
>
> | Dataset       | Proto Dist. (HXE) | Proto Dist. (XE) |
> |---------------|:----------------: |:----------------:|
> | ppastoris     |    **0.676**      |   0.666          |
> | scerevisiae   |    **0.434**      |     0.342        |
> | mRFP          |    **0.819**      |     0.752        |
> | E. Coli       |    **50.8**       |     48.6         |
> | Fungal        |    0.735          |     **0.740**    |
> | COVID-19      |    **0.785**      |     0.775        |
>
> **The base point of the exponential map.**
> This choice of base point for the exponential map is well-established in the hyperbolic learning literature [2–8], as it greatly simplifies the exponential map and has been shown empirically to have barely any impact on performance [4]. Moreover, in our setting the origin has an important geometric meaning. Namely, it represents the origin of the tree. Therefore, setting the base point to the origin essentially centers token representations around the root of the tree, which intuitively seems to be a sensible approach.
>
> To test reviewer’s hypothesis if origin-centered exponential map creates bottleneck, we ran experiments using a learnable base point with fixed entailment prototypes. Our origin choice performs better on 4/9 datasets and on par on 4/9, validating our initial design (Appendix L).
>
> | Dataset | Learnable | Origin |
> |---------|-----------|--------|
> | Ab1 | 0.701 |  **0.751** |
> | scerevisiae | 0.369 |  **0.397** |
> | COVID-19 | 0.783 |  **0.807** |
> | Fungal | 0.724 |  **0.741** |
> | EColi |  **50.6** | 48.4 |
> | dmelanogaster |  **0.451** |  **0.450** |
> | Ab2 | **0.570** |  **0.569** |
> | mRFP | **0.805** |  **0.802** |
> | ppastoris |  **0.671** |  **0.671** |

---

> ### Author Response · Authors · 2025-11-20
> **Response to review- Response 2/4**
>
> **The poor performance of MLR and the role of hyperbolic geometry.**
> As the reviewer notes, the MLR variant performs poorly, which underlines that simply switching to hyperbolic geometry is not enough. Our hypothesis regarding this observation is most evident when tracking hyperbolic norms of representations during training (added Figure 5 in Appendix Section F). Essentially, standard MLR tends to drive the embeddings away from the origin, resulting in quickly growing hyperbolic norms of the representations [9]. In hyperbolic space, large norms quickly lead to numerical instability, and hyperbolic learning libraries such as HypLL handle this by clipping points back into a bounded region. As training progresses, MLR therefore produces heavily clipped representations, effectively discarding information encoded in the norm and hindering learning. The hyperbolic prototypes approach, however, is not prone to ever-growing representation norms and hence requires no clipping, which leads to increased representation flexibility and stability. We have added the detailed explanation in Appendix F.
>
>
> **Missing values for CodonBERT.** The missing CodonBERT results for some of the datasets stem from difficulties in running the publicly released model on these inputs. In our experiments, we encountered previously reported discrepancies documented in the CodonBERT GitHub issues (https://github.com/Sanofi-Public/CodonBERT/issues/7). For these datasets, the tokenizer produced identical embeddings regardless of the input sequence, rendering the resulting representations unusable for downstream evaluation. The underlying issue, first reported in an open GitHub ticket (Issue #7), has remained unresolved for over a year.
>
>
> **Comparison to EVO 2.**
> We thank the reviewer for their suggestion. Based on their and the other reviewer's suggestions, we have included several additional baselines, namely EVO 2 (7B), Helix-mRNA, mRNA-FM, and AIDO.RNA-CDS. The results are shown in the table below. Due to the computational and time constraints, we are not able to obtain results of all models on all downstream datasets. As can be seen, our model outperforms the additional baselines, including EVO 2, on all of the datasets except for AIDO.RNA-CDS on COVID-19, where our model performs on par, and mRNA-FM on _E. coli_ where mRNA-FM is the strongest model. These additional results have been included in the paper. Missing value for Fungal dataset with AIDO.RNA-CDS indicates the runs did not execute due to hardware limitations. We have also included these in Appendix Section G.
>
>
> | Dataset       | Helix-mRNA |  mRNA-FM | EVO 2 | AIDO.RNA-CDS | Proto Dist. (Ours)     | Proto Entail. (Ours)|
> |---------------|:----------:|:--------:|:-----:|:------------:|:----------------------:|:------------------------:|
> | Ab1           |    0.535   |   0.656  | 0.129 |     0.663    |    0.713               |   **0.751**              |
> | Ab2           |    0.283   |   0.373  | 0.141 |     0.398    |  **0.575**             |     0.569                |
> | mRFP          |    0.432   |   0.739  | 0.239 |     0.787    |  **0.819**             |     0.802                |
> | COVID-19          |    0.643   |   0.762  | 0.386 |     0.804    |    0.785               |   **0.807**              |
> | Fungal        |    0.689   |   0.722  | 0.400 |              |    0.735               |   **0.741**              |
> | Ecoli         |    40.0    | **53.3** |  40.0 |     52.9         |     50.8               |      48.4                |

---

> ### Author Response · Authors · 2025-11-20
> **Response to review- Response 3/4**
>
> **The improvement on long sequences.**
> The benefits of hyperbolic space in OOD settings have been observed empirically in many works [2, 6, 10-12], but, to the best of our knowledge, a clear explanation of this effect has not been found yet. We have a hypothesis regarding this commonly observed effect which we will try to formalize using some strong assumptions here. Our goal is not to form a rigorous proof, but to provide an intuition as to why we believe this to indeed be a result of hyperbolic geometry.
>
> For this discussion, we will assume that:
> 1. We have either a Euclidean or hyperbolic model with a representation space in which instances of each class $1, \ldots, C$ are contained to a bounded ball $B_i^M = \\{x \in M : d_M(x, c_i) < r_i\\}$, where $M \in \\{ \mathbb{R}^n, \mathbb{D}^n \\}$;
> 2. These balls have equal radii ($\forall i, j: r_i = r_j$) and their centers have equal distance from the origin ($\forall i, j:||c_i||_M = ||c_j||_M$);
> 3. The balls containg each class' representations do not overlap ($\forall i, j: B_i^M \cap B_j^M = \emptyset$), so the model can properly separate classes;
> 4. When the model attempts to represent an instance of class $i$ from an out-of-distribution sample, such as a long sequence, the result is contained to a ball $\tilde{B}_i^M$ centered at the same $c_i$, but for some radius $\tilde{r}_i \geq r_i$, where the size of the difference between the 2 radii is determined by the specific distribution shift.
>
> Under these assumptions, OOD samples become problematic for the model, once the $\tilde{r}\_{i}$'s become large enough that the $\tilde{B}\_{i}^M$'s start to overlap, since the model will then no longer be able to perfectly discern between instances of all classes. In hyperbolic space, these $\tilde{r}\_{i}$'s can be larger than in Euclidean space, which is somewhat easily shown through an example.
>
> Suppose we have 4 classes, in 2-dimensional hyperbolic or Euclidean space and suppose that for each class $i$ we have evenly placed centers such that $||c\_{i}||\_{M} = 2$, with $r_i = 1$. For instance, in $\mathbb{R}^2$ we could have $c_1 = (2, 0)$, $c_2 = (0, 2)$, $c_3 = (-2, 0)$ and $c_4 = (0, -2)$ or in the Poincaré disk $\mathbb{D}\_{1}^2$ we could similarly have $c_1 = (\tanh(1), 0)$, $c_2 = (0, \tanh(1))$, $c_3 = (-\tanh(1), 0)$ and $c_4 = (0, -\tanh(1))$. Now suppose we have some distribution shift such that only $\tilde{r}\_{1}$ is larger than $r_1$. Then, you can compute how large the difference $\tilde{r}\_{1} - r\_{1}$ can become before the model can no longer distinguish perfectly between classes. For the Euclidean case, this is $d_{\mathbb{R}}(c_1, c_2) - (r_1 + r_2) \approx 0.828$, while for the hyperbolic case we get $d_{\mathbb{D}} (c_1, c_2) - (r_1 + r_2) \approx 1.34$. This shows that the hyperbolic model, with a similar organization of its representation space but a different geometry, is robust to larger distribution shifts. As the representations cover larger areas of the space, this difference becomes more pronounced due to the increased effect of the negative curvature in hyperbolic space. We hope that this artifical example can provide some intuition as to why we believe that the OOD improvements can be attributed to the hyperbolic geometry.

---

> ### Author Response · Authors · 2025-11-20
> **Response to review- Response 4/4**
>
> We appreciate the reviewer's thoughtful and constructive feedback and hope that our detailed responses have resolved the noted issues. Should the reviewer find the revisions satisfactory, we would be grateful if they could raise their score accordingly.
>
> **References**
>
> [1] Bonnabel, Silvere. "Stochastic gradient descent on Riemannian manifolds." IEEE Transactions on Automatic Control 58.9 (2013): 2217-2229.
>
> [2] Mettes, Pascal, et al. "Hyperbolic deep learning in computer vision: A survey." International Journal of Computer Vision 132.9 (2024): 3484-3508.
>
> [3] Khrulkov, Valentin, et al. "Hyperbolic image embeddings." Proceedings of the IEEE/CVF conference on computer vision and pattern recognition. 2020.
>
> [4] Ermolov, Aleksandr, et al. "Hyperbolic vision transformers: Combining improvements in metric learning." Proceedings of the IEEE/CVF conference on computer vision and pattern recognition. 2022.
>
> [5] Atigh, Mina Ghadimi, et al. "Hyperbolic image segmentation." Proceedings of the IEEE/CVF conference on computer vision and pattern recognition. 2022.
>
> [6] Van Spengler, Max, Erwin Berkhout, and Pascal Mettes. "Poincare resnet." Proceedings of the IEEE/CVF International Conference on Computer Vision. 2023.
>
> [7] Desai, Karan, et al. "Hyperbolic image-text representations." International Conference on Machine Learning. PMLR, 2023.
>
> [8] Pal, Avik, et al. "Compositional Entailment Learning for Hyperbolic Vision-Language Models." The Thirteenth International Conference on Learning Representations.
>
> [9] Ganea, Octavian, Gary Bécigneul, and Thomas Hofmann. "Hyperbolic neural networks." Advances in neural information processing systems 31 (2018).
>
> [10] Kasarla, Tejaswi, Max van Spengler, and Pascal Mettes. "Balanced Hyperbolic Embeddings Are Natural Out-of-Distribution Detectors." arXiv preprint arXiv:2506.10146 (2025).
>
> [11] Hong, Jie, et al. "Curved geometric networks for visual anomaly recognition." IEEE transactions on neural networks and learning systems (2023).
>
> [12] Guo, Yunhui, et al. "Clipped hyperbolic classifiers are super-hyperbolic classifiers." Proceedings of the IEEE/CVF Conference on Computer Vision and Pattern Recognition. 2022.

---

> > ### Author Response · Authors · 2025-11-26
> > **Friendly request for feedback...**
> >
> > Dear reviewer,
> >
> > We kindly ask for a feedback regarding our new experiments, detailed explanations and revised version of the paper. We would like to understand if your concerns are now adequately addressed and if you would kindly consider raising your score in light of this. Additionally, if you would like to voice additional concerns or add details to any of the discussed points, we would be delighted to incorporate them in the paper in adequate form as well. We want to thank you in advance for your time and efforts!

---

### Author Response · Authors · 2025-12-02
**Summary of Reviews and Evidence of Resolution for AC's consideration**

Given the increased load on ACs during the discussion phase, we provide an expanded yet concise summary of the reviews and the discussion, highlighting the main points of agreement, criticism, and our responses.

Across reviews, there was broad agreement on three strengths of the work. First, reviewers emphasized the **novelty, motivation, and clarity** with which the proposed methodology was presented. Second, they highlighted the **strong biological grounding** of the method and the effective fusion of **Euclidean and hyperbolic components**. Third, they noted the **comprehensive empirical evaluation**, which demonstrated clear advantages of our model, both in predictive performance and in robustness/generalizability across varied datasets.

Below we summarize the major concerns raised and how they were addressed in our responses and revisions.
---

### 1) Fixed prototypes
The prototypes in the original paper were fixed. Reviewers **bM9n** and **Ak6U** suggested that learnable prototypes might improve our model performance further. \
**Response:** We ran experiments using **learnable prototypes**, which improved results further for our model in several cases. The paper was revised to present both fixed and learnable options, emphasizing that prototype learning is beneficial.

### 2) Additional ablations
Reviewer **bM9n** requested ablations on (1) the loss function (standard vs. hierarchical cross-entropy) and (2) the base-point for the exponential map. \
**Response:** We provided ablation evidence showing that our **chosen loss and base-point yield superior performance**, validating our design choices.

### 3) Validity of codon-level tokenization
Reviewer **Vc4D** questioned the correctness of codon-level tokenization, based on an incorrect assumption about the data scope. \
**Response:** We clarified that we exclusively model **CDS regions**, where codon-level tokenization is appropriate and data is annotated at codon level. After clarification, the reviewer acknowledged resolution, and the paper now clearly states this data scope.

### 4) Additional mRNA foundation baselines
Reviewers **Vc4D** and **5EoM** requested results with more mRNA Foundation Model (FM) baselines.
**Response:** We added results for most downstream datasets with multiple baseline FMs namely, **mRNA-FM, AIDO.RNA-CDS, Helix-mRNA, and EVO 2**. Our model **achieves the strongest performance**, despite massive scale of these baselines (**5x-140x larger than our model**).

### 5) Reviewer **5EoM** expressed concern about code availability and reliance on prior dataset references.
**Response:** We clarified that **all code and pretrained weights will be released upon publication**, and we expanded dataset descriptions to ensure the paper is **self-contained and reproducible**. We also mentioned that datasets and splits were obtained from prior works for comparability.

### 6) Novelty
Reviewer **5EoM** questioned the novelty of the approach. \
**Response:** To our knowledge, **HyperHELM is the first method to perform language modeling with hyperbolic prototypes**, beyond mRNA applications. We combine prototype learning with **hyperbolic entailment cones** and **HS-DTE** for biologically grounded hierarchical structure.  Importantly, simply replacing the geometry of HELM by Euclidean geometry leads to the hyperbolic MLR method, which performs very poorly, as shown in Table 1 of the main text.
Our paper was submitted to the **ICLR 2026 physical sciences track**, which encourages principled methods tailored to scientific domains; our empirical results demonstrate such benefits.

### 7) Performance variance estimation
Reviewers **Vc4D** and **5EoM** suggested reporting variance. \
**Response:** Because we rely on **fixed splits from prior work**, variance estimation is not possible without creating new splits, which would reduce comparability and impose large computational demands. On established benchmarks, results already show **consistent, substantial improvements**:  **+13.3% (Ab1), +51.5% (Ab2), +7.7% (mRFP), +2.2% (COVID-19), +4.4% (Fungal), +4.8% (iCodon), and +7.1% over EVO-2 on E. coli**.  Gains are large, systematic, and span heterogeneous datasets. Our model also outperforms much larger baselines, despite being **5×–140× smaller**, strengthening the evidence for methodological value.

---
We also addressed all remaining minor concerns.

**Score update during rebuttal and discussion:** After responses, reviewer Vc4D increased score; Ak6U had initial positive assessment which they maintained, bM9n did not participate in discussion, while 5EoM asked some final questions, which we addressed but the discussion period ended abruptly due to current circumstances before they could update their score!

**We request the AC to consider our thorough revisions, new experiments, which earned praise and led to score increases in their final evaluation.**

Kind regards, \
The authors

---

### Meta-Review · Area_Chair_SJPd · 2026-01-07

**Summary:**

Reviewers found the paper to be a well-motivated and biologically grounded application of hyperbolic geometry to mRNA language modeling, with generally strong empirical performance. However, several reviewers raised concerns regarding biological scope and clarity, baseline completeness, reproducibility, variance estimation, and the primarily application-level novelty of the contribution.

**Reviewer Concerns:**

Reviewers raised concerns about the biological validity and clarity of the modeling scope (specifically the CDS-only focus versus full mRNA claims), completeness and appropriateness of baseline comparisons, lack of variance estimates under fixed dataset splits, reproducibility details (including dataset splits and release plans), and the limited methodological novelty beyond adapting an existing hierarchical framework to hyperbolic space. The rebuttal addressed many points through additional ablations, new foundation-model and supervised baselines, clarifications regarding CDS alignment and reading frames, expanded related work, and revisions to make the CDS-only scope explicit. However, some concerns remain only partially resolved: performance variance could not be quantified, evaluation still relies heavily on fixed splits, and the contribution remains primarily applied rather than methodological. As a result, despite substantial improvements, not all reviewer concerns are fully alleviated.

**Reviewer Scores:**

Reviewer scores initially spanned from clearly negative to marginally above the acceptance threshold. Following the rebuttal, at least one reviewer explicitly raised their score after many concerns were addressed, while others remained cautious due to remaining issues around evaluation rigor and scope. Overall, while reviewer confidence increased, the score distribution would likely remain mixed, with improvements insufficient to clearly move the paper above the acceptance threshold.

---

### Decision · Program_Chairs · 2026-01-26

Reject